# Scalable synthesis of phosphorescent SiO₂ nanospheres and their use for angle-dependent and thermoresponsive photonic gels with multimode luminescence

Changxing Wang ⓘ, Yayun Ning, Yifan Yue, Guoli Du ⓘ, Yuechi Xie, Jianing Li, Nazia Bibi, Xiaoxiang Wen, Jianing Li, Sen Yang ⓘ ✉ & Xuegang Lu ⓘ ✉

Developing room-temperature phosphorescent (RTP) materials with micro-scale periodic structures presents a promising prospect for future optical applications but remains challenging due to the complex integration of luminescent and structural components. Herein, we present a strategy for large-scale production of RTP silica nanospheres (RTP SiO₂ NPs) with a low dispersity in size using a modified Stöber method, where organic molecules are embedded in silica networks and subsequently undergo in-situ carboni-zation, aggregation and crystallization to form phosphorescent carbon dots under high-temperature calcination. These NPs can self-assemble into pho-tonic crystal (PC) structures, enabling the straightforward integration of structural color, fluorescence (FL) and RTP to achieve multimodal luminescent properties. The angle-dependent photonic bandgap (PBG) generated by the physical periodic structure modulates light propagation in RTP PC gels, creating FL and RTP angle-dependent chromatic responses. Temperature-induced refractive index changes between SiO₂ and the liquid matrix further enable dynamic control of light-scattering states, significantly altering trans-mittance and emission intensities of FL and RTP. This fusion of physical pho-tonic structures with luminescence offers potential approach for constructing advanced multimodal luminescent devices.

Room-temperature phosphorescence (RTP) materials, characterized by long-lived emission and low energy dissipation, offer transforma-tive potential across optoelectronic applications including sensing, displaying, decoration, information encryption, and environmental monitoring[1–6]. Recent advances in nanotechnology and programmable self-assembly have driven a paradigm shift in RTP modulation, from manipulating complex chemical structures to tuning robust physical microstructures[7,8]. However, conventional RTP materials often face challenges in the precise design of physical morphology and structure, especially in the large-scale production, where the morphology and structure of RTP materials are difficult to maintain consistency. By precisely controlling the physical structure and morphology of phos-phorescent materials, not only can alternative light signals be gener-ated through modulation of physical microstructures, such as transmitted light, scattered light, and reflected light, to achieve mul-timode luminescence, but potential interactions between different light signals can also be realized[9]. For example, integrating RTP materials with periodic nanostructures allows the formation of a

School of Physics, MOE Key Laboratory for Nonequilibrium Synthesis and Modulation of Condensed Matter, Shaanxi Province Key Laboratory of Advanced Functional Materials and Mesoscopic Physics, Xi'an Jiaotong University, Xi'an, China. ✉e-mail: yangsen@mail.xjtu.edu.cn; xglu@mail.xjtu.edu.cn

photonic bandgap (PBG), which can modulate the propagation of the phosphorescence itself[10]. Therefore, developing cost-effective, high-throughput, and scalable manufacturing strategies of phosphorescent materials while ensuring the uniformity and regularity of the physical structures is the key issue in developing RTP materials with intrinsic physical light modulation capabilities.

Furthermore, when the external environment such as light, temperature, electric fields and magnetic fields changes, stimuli-responsive chromic materials (SRCM) undergo reversible color changes, which has garnered significant research interest in the field of smart materials[11–15]. Typically, the crystal structure, physical arrangement, and chemical properties of these materials change under stimulation, hereby altering their light absorption, reflection, transmission, and emission performances. Despite the transformative potential of SRCM across multidisciplinary applications, critical challenges are still remained in their functional implementation. On one hand, traditional SRCM typically require external energy input, exhibiting high response lag and short emission lifetimes[16,17]. On the other hand, the single optical expressions of most SRCM fail to achieve precise matching of optical signals with physical or chemical parameters in complex environments[18,19]. Theoretically, by rationally designing and integrating the stimulus matrix, physical structure, and optical modulation units, SRCM with multi-dimensional optical responses can be realized[20,21]. Independent or synergistic responses between different optical signals can not only provide rich visual effects and high conversion efficiency, but also improve response accuracy and sensitivity. However, to date, the proposed design and

manufacturing strategies for multimode integrated stimulus-responsive optical devices are very limited. Therefore, efficient design of optical devices with high optical freedom and customized response remains a significant challenge.

To address the above challenges, we have developed a strategy for preparing RTP SiO$_2$ nanoparticles (RTP SiO$_2$ NPs) with uniform spherical structures, by calcining SiO$_2$ NPs encapsulating organic low-molecular-weight molecules (Fig. 1a). During the calcination process, the embedded organic molecules in SiO$_2$ matrix undergo in-situ formation of fluorescent carbon dots (FL CDs) through carbonization, aggregation and crystallization, while the covalent C-Si bond network between the CDs and SiO$_2$ matrix is gradually built up, stabilizing the triplet excited state to produce long-lived RTP emission. Notably, as shown in Fig. 1b, this strategy is applicable to various incorporated organic molecules, reducing preparation costs, improving operational convenience, and enabling scalable production of RTP SiO$_2$ NPs (>700 g per batch). As-prepared RTP SiO$_2$ NPs maintain their original self-assembly ability, and the resulting photonic crystals (PCs) through simple evaporation-induced self-assembly exhibit multimodal luminescent properties, including structural color (SC), FL and RTP. More intriguingly, photonic gel (PC gel) assembled from RTP SiO$_2$ NPs, relying on the PBG generated by their periodic microstructures, can regulate the propagation state of FL and RTP, thus achieving cooperative interactions of multiple optical signals (Fig. 1c). Due to the angle dependence of the PBG, the photon resonance peaks of FL and RTP also show angle-dependent behavior. Therefore, by combining the static FL and RTP emissions with the dynamic photonic resonance

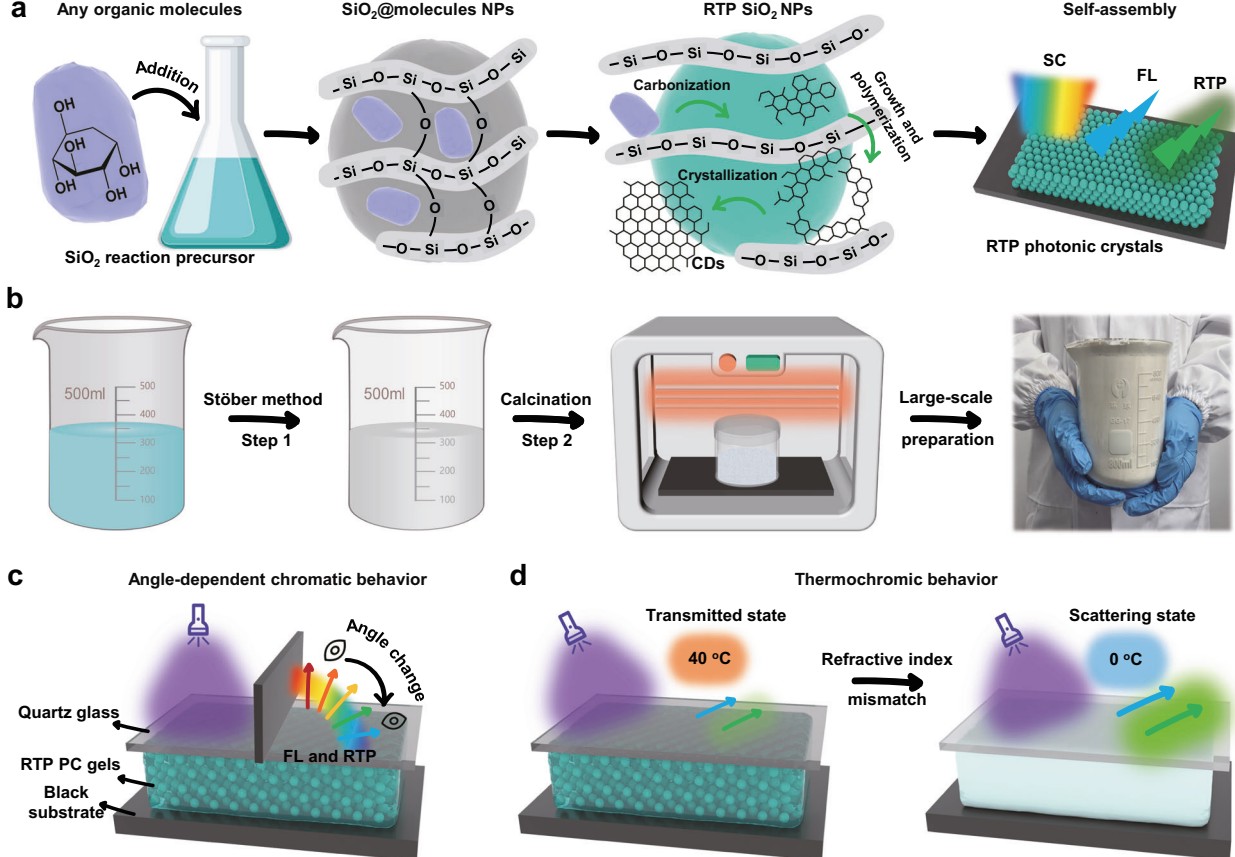

**Fig. 1 | Schematic illustration of preparation process of RTP SiO$_2$ NPs and multi-response chromatic behavior of photonic gels. a** Schematic depiction of the design process for preparing spherical RTP SiO$_2$ NPs and self-assembled multimode luminescent PCs, the gray and green spheres represent the SiO$_2$ NPs doped with organic molecules before and after calcination, respectively. **b** Schematic diagram

of large-scale preparation process for RTP SiO$_2$ NPs. **c** Schematic illustration of angle-dependent chromatic behavior induced by photon bandgap. **d** Schematic illustration of thermal-induced self-scattering enhanced luminescence properties of PC gels.

peaks, an interesting angle-dependent chromatic phenomenon can be realized. Furthermore, due to the differing temperature sensitivities of the refractive index of RTP SiO₂ NPs and liquid matrix, the PC gel demonstrates a temperature dependent dynamic refractive index matching characteristic (Fig. 1d). As the temperature decreases, the originally matched refractive index of the two phases gradually tends to mismatch, resulting in the PC gel changing from transparent state to white scattering state, and the enhanced light scattering ability significantly enhances the emission intensities of FL and RTP. Clearly, by combining physical structures with photoluminescence, our findings provide a feasible approach for constructing optical devices with multi-stimulus response and customized optical signal expression.

## Results

### Optical and self-assembly properties of RTP SiO₂ NPs

The preliminary molecular-doped SiO₂ NPs are prepared by adding glucose molecules to the precursor during the traditional Stöber method, followed by calcination at 575 °C to obtain RTP SiO₂ NPs. As shown in Fig. 2a, the transmission electron microscope (TEM) image of RTP SiO₂ NPs shows a low dispersity in size, homogeneous and regular spherical morphology with average particle size of 284 nm, and the morphology and particle size are insusceptible in calcination (Supplementary Figs. 1, 2). Energy dispersive spectrometer (EDS) elemental mapping images clearly display that there are only C, O and Si elements in RTP SiO₂ NPs, and the uniform distribution of C elements within the spherical regions, indicating that glucose molecules are encapsulated in SiO₂ NPs (Fig. 2b and Supplementary Fig. 3). Furthermore, RTP SiO₂ NPs with particle diameters ranging from 206 to 284 nm can be easily prepared by adjusting the ethanol content during the reaction (Supplementary Fig. 4).

As illustrated in Fig. 2c, under excitation at 365 nm UV light, the RTP SiO₂ NPs display blue FL emission at 466 nm, while the green RTP emission band is centered at 504 nm. Whether FL or RTP, RTP SiO₂ NPs exhibit strong excitation dependent emission behavior, covering almost the entire visible band (Fig. 2d and Supplementary Fig. 5). After exposure to different excitation light sources, even the white light, RTP SiO₂ NPs present colorful afterglow, and the afterglow persisting is over 17 s at 365 nm excitation (Supplementary Fig. 6 and Supplementary Movie 1). We preliminarily speculate that this excitation dependent optical behavior may be caused by different defective luminescent states formed by organic molecules or generation of different fluorescent molecules during the calcination process[22,23]. Besides, the time-resolved RTP emission spectra of RTP SiO₂ NPs excited at 365 nm further confirms the long-lived RTP emission with an optimum average decay lifetime ($\tau_{av}$) of 2 s at the 504 nm emission band (Fig. 2e). Furthermore, as the temperature increasing from 77 to 347 K, a noticeable decrease in RTP emission intensity and RTP decay lifetime is observed, confirming the absence of a thermally enhanced process, and presenting typical phosphorescence characteristics (Supplementary Fig. 7 and Supplementary Table 1)[24,25]. In addition, the optical intensity of RTP SiO₂ exhibits good stability in various organic solvents and most metal ion solutions (Supplementary Figs. 8, 9), which reflects a good anti-quenching ability, and further proves that the photoluminescence center of RTP SiO₂ NPs is located in the interior of SiO₂ NPs.

Interestingly, by simple evaporation-induced self-assembly, RTP SiO₂ NPs can be arranged in periodic array structures. As presented in Fig. 2f, g, the RTP SiO₂ NPs exhibit an ordered close-packed state with two or more layers, respectively, which is attributed to the (111) plane of the face-centered cubic (fcc). Notably, we also observed the characteristic arrangement pattern of the (100) planes of the fcc structure (Supplementary Fig. 10). In addition, scanning electron microscopy (SEM) has also confirmed that the synthesized RTP SiO₂ NPs are capable of forming highly ordered, closely packed periodic structures, which exhibit the characteristics of PC structure

(Fig. 2h)[26]. Therefore, the multimodal PC assembled from RTP SiO₂ NPs not only retain its intrinsic FL and RTP emission capabilities, but also possess the ability to manipulate light through its physical microstructures, thereby achieving vivid structural colors. As shown in Fig. 2i, the assembled multimodal PC displays angle-dependent structural colors, bright blue FL and long-persistent green RTP under different light stimulation (Supplementary Movie 2). As the observation angle ($\theta$) decreases from 90° to 20°, the structural colors of the multimodal PC gradually transition from red to blue, which is consistent with the angle-dependent reflection spectra (Fig. 2j). This phenomenon is in accordance with Bragg's law for traditional PC, as described by Eq. (1)[27]:

$$\lambda \propto d \sin\theta \qquad (1)$$

where $d$ refers to the distance between the adjacent centers of the RTP SiO₂ NPs, and $\theta$ is the angle between the PCs surface and the incident light or reflected light. Moreover, the multimodal PC self-assembled from RTP SiO₂ NPs with varying diameters exhibit distinct structural colors (Supplementary Figs. 11-13), whereas FL and RTP emission exhibit diameter-independent properties and the persistence of afterglow is more than 9 s (Supplementary Figs. 14, 15).

### Mechanism investigations of RTP SiO₂ NPs production

Generally, the realization of phosphorescent materials needs to meet two basic requirements, one is the chemically defective luminescence centers, the other is a rigid environment to stabilize the triplet excitons in these luminescent centers[28]. Therefore, we first explore the formation of chemically luminescent centers within RTP SiO₂ NPs during the calcination process. Compared with the uncalcined SiO₂ NPs, a large number of structural defects are generated inside the calcined particles, and these defects have obvious lattice fringes with a lattice spacing of approximately 0.21 nm (Fig. 3a). Upon etching away the silica network with hydrofluoric acid, the exposed internal defect structure is revealed to be composed of spherical nanoparticles and retains similar lattice spacing, exhibiting the characteristic fluorescent CDs structure[29,30]. Hence, it is obvious that during the calcination process, glucose molecules undergo in-situ carbonization, aggregation, and crystallization within the SiO₂ NPs to generate CDs. Furthermore, the formation and growth process of CDs is studied by calcining SiO₂ NPs doped with glucose molecules at different temperatures. It is found that the calcination temperature of 325 °C is sufficient to convert the glucose molecules inside the SiO₂ NPs into CDs (Supplementary Fig. 16), and the size of CDs increases with increasing calcination temperature, growing from 2.82 nm at 325 °C to 7.47 nm at 575 °C (Fig. 3b). Moreover, the RTP spectra and time-resolved RTP spectra of RTP SiO₂ NPs calcined at different temperatures show optimum intensity and the longest RTP lifetime at 575 °C calcination (Supplementary Figs. 17, 18, and Supplementary Table 2). It can be inferred that as the calcination temperature increases (from 325 to 575 °C), the crystalline structure of the CDs derived from glucose becomes more stable, and the non-radiative transitions of electrons is weakened, leading to a higher overall luminescence efficiency (Supplementary Figs. 19, 20, and Supplementary Table 3)[31]. High-resolution TEM (HR-TEM) has validated this point. As presented in Fig. 3c, the generated CDs exhibit characteristics of small size and low crystallinity at 325 °C calcination temperature, and these small-sized CDs grow and simultaneously aggregate as the temperature increases to 425 °C, forming larger-sized CDs with high lattice defects. As the temperature further increases to 575 °C, the defect structures within the CDs are gradually filled and reduced, resulting in the formation of large-sized CDs with high crystallinity. It is worth mentioning that the captured arrangement of carbon atoms deviates from the hexagonal honeycomb structure, which may be attributed to the varying spatial orientations of the CDs[32]. The corresponding intensity ratio between crystalline G

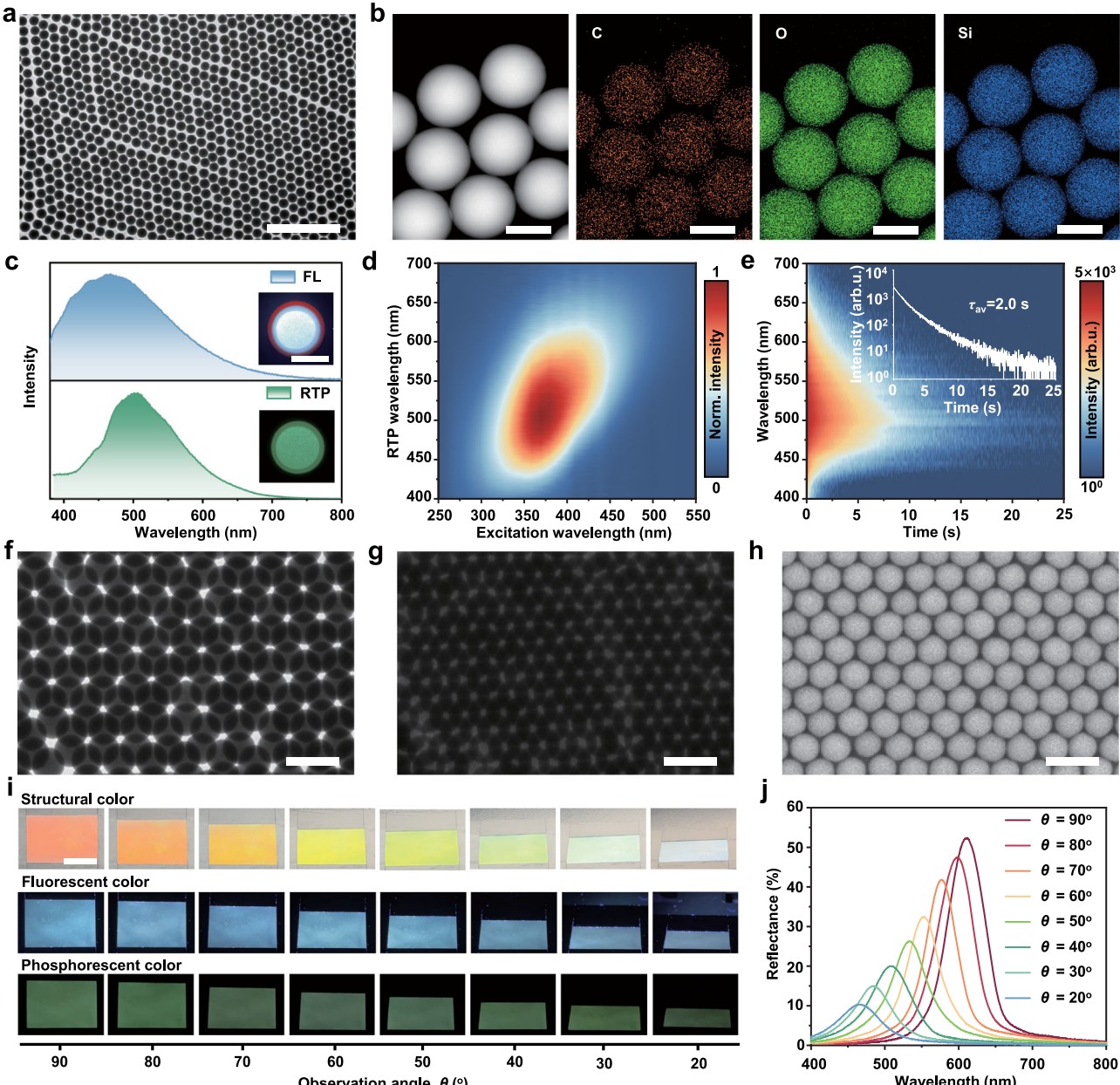

**Fig. 2 | Morphology characterization, optical properties and self-assembly behavior of RTP SiO₂ NPs. a** TEM image of RTP SiO₂ NPs (scale bar: 2 μm). **b** High-angle annular dark-field scanning transmission electron microscopy (HAADF-STEM) and EDS elemental mapping images of RTP SiO₂ NPs (scale bar: 200 nm), where the red, green, and blue represent the elemental distribution of C, O and Si, respectively. **c** FL (blue) and RTP (green) spectra of RTP SiO₂ NPs under 365 nm excitation wavelength. Insets show photographs of the RTP SiO₂ NPs under 365 nm UV excitation lamp on and off, scale bar: 5 cm. **d** Excitation-phosphorescence mapping of the aqueous RTP SiO₂ NPs at room temperature. **e** Time-resolved emission spectra of RTP SiO₂ NPs under 365 nm excitation wavelength, the inset image is lifetime decay profile of the RTP for RTP SiO₂ NPs under 365 nm excitation wavelength. **f** and **g** TEM images of RTP SiO₂ NPs in a double-layer and triple-layer close-packed structure, respectively (scale bar: 500 nm). **h** SEM image of the closely packed RTP SiO₂ NPs (scale bar: 500 nm). **i** The photographs of the angle-dependent structural colors, FL and RTP of the multimodal PCs under daylight, 365 nm UV lamp on and off, respectively, scale bar: 1 cm. **j** Angle-dependent reflection spectra of the multimodal PCs self-assembled by RTP SiO₂ NPs with diameter of 284 nm, from red to blue represents the θ gradually decreasing from 90° to 20°. Source data are provided as a Source Data file.

band ($I_G$) and disordered D band ($I_D$) in Raman spectra increases from 0.827 to 1.111 with the calcination temperature rising from 325 to 575 °C, which is consistent with the HR-TEM results (Supplementary Fig. 21). Consequently, the photoluminescence centers of RTP SiO₂ NPs after calcination are derived from internal in-situ generated CDs, and further confirms that even a single organic low-molecular-weight molecule carbon source can also be converted to CDs through car-bonation, aggregation and crystallization within the silica matrix (Fig. 3d).

Subsequently, we analyze the reasons for the formation of stable triplet states ($T_1$) responsible for the long-lived RTP of CDs during the calcination process. The temperature-dependent Fourier transform-infrared spectroscopy (FT-IR) show a gradual increase in the C-Si bonds at approximately 807 cm⁻¹ with increasing calcination temperature, while the Si-O-Si/Si-O-C groups at approximately 1046–1299 cm⁻¹ gradually decrease, indicating that during the calci-nation process, some Si-O-Si/Si-O-C bonds break and form new C-Si covalent bonds with the CDs (Supplementary Fig. 22). Besides, the

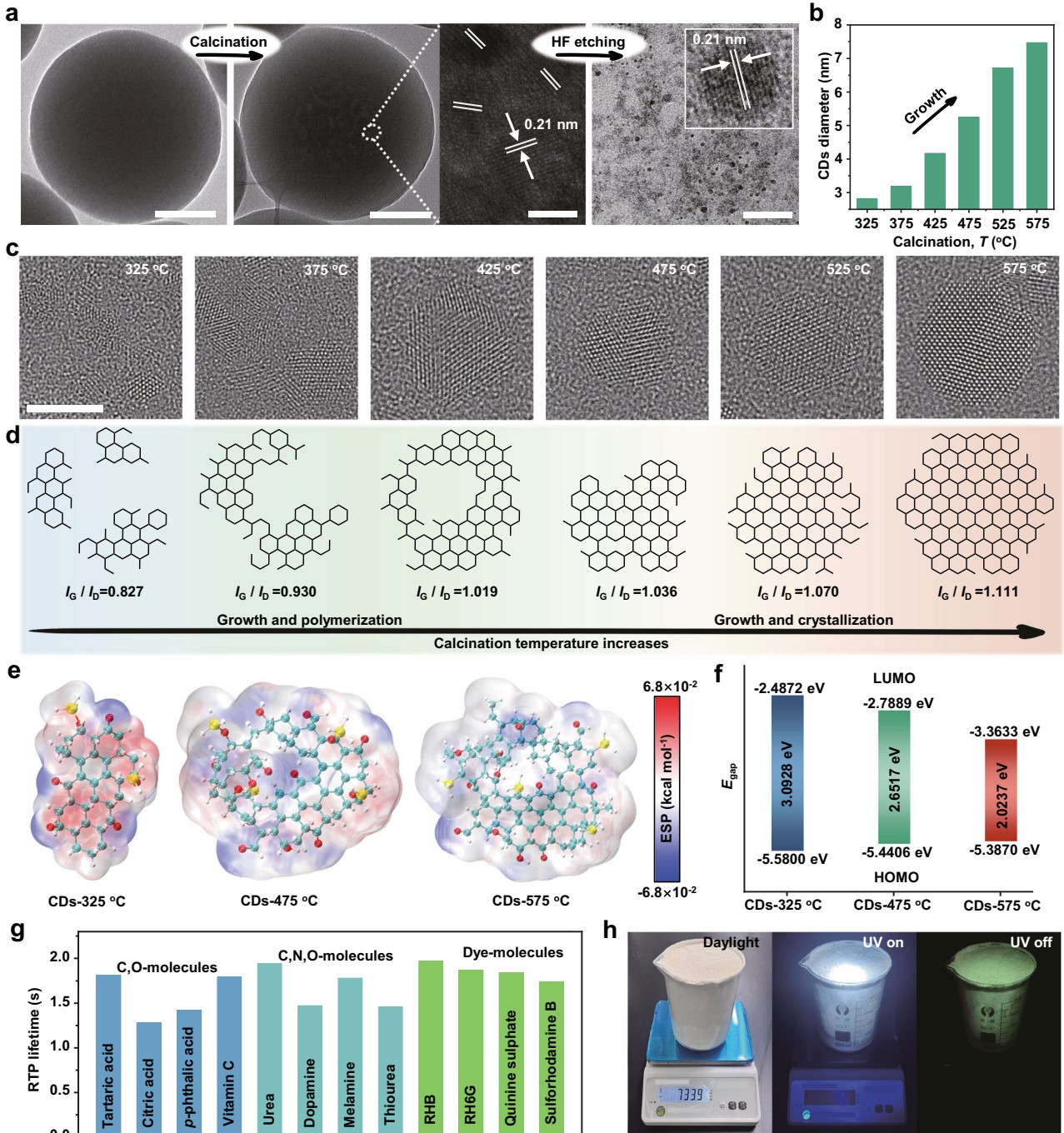

**Fig. 3 | RTP mechanism of RTP SiO₂ NPs, the universality of strategy and large-scale preparation. a** TEM images of SiO₂ NPs doped with glucose molecules before calcination, after calcination and after hydrofluoric acid etching, the scale bars from left to right are 100 nm, 100 nm, 5 nm and 50 nm, respectively. **b** The relationship between diameter of CDs formed inside RTP SiO₂ NPs and calcination temperature. **c** HR-TEM images of CDs growth inside SiO₂ NPs at different calcination temperatures, the scale bars are 5 nm. **d** Schematic diagram of the growth process of CDs inside the SiO₂ NPs during the calcination process. **e** The molecular surface electrostatic potential for the molecular structure of CDs at 325, 475, and 575 ℃ calcination conditions designed by HR-TEM images. The red region, blue region, and white region indicate the lower electron density, densely electron-dense region, and neutrality, respectively. The cyan, gray, red, and yellow spheres in the molecular structure represent carbon, hydrogen, oxygen, and silicon atoms, respectively. The ESP is electrostatic potential of the CDs. **f** The calculated energy gaps of HOMO and LUMO for the corresponding molecular structure of CDs. **g** The RTP lifetime of RTP SiO₂ NPs synthesized by doping various organic small molecules, where C, O and N represent the carbon, oxygen, nitrogen elemental composition of the corresponding organic molecules, respectively, the blue, cyan, and green bar charts represent different types of organic molecules. **h** Photographs of large-scale prepared RTP SiO₂ NPs under sunlight, UV radiation, and after UV shut-off. Source data are provided as a Source Data file.

fitted high-resolution Si $2p$ results spectra of RTP SiO₂ NPs show an increase in the relative content of C-Si bonds at 102.1 eV as increasing calcination temperature (Supplementary Fig. 23). Notably, the C $1s$ and O $1s$ band of RTP SiO₂ NPs verify the presence of O-related defect states, which provide a guarantee for long-lived RTP (Supplementary Figs. 24–27). Taken together, during the calcination process, the growth and crystallization of CDs are accompanied by a simultaneous formation of a rigid covalent C-Si network with SiO₂ matrix, which

facilitates the stabilization of triplet excitons and effectively suppresses the non-radiative transitions of the $T_1$, thereby generating long-lived RTP emission[33,34].

Theoretical calculations based on the distribution of molecular surface electrostatic potential (ESP) and density functional theory (DFT) are carried out to further reveal the mechanism of RTP property from RTP $SiO_2$ NPs. Based on HR-TEM and FT-IR results, the molecular structure of CDs at 325, 475 and 575 °C calcination conditions are selected and designed as computational models (Supplementary Fig. 28). With the increase of calcination temperature, the ESP of the CDs inside the RTP $SiO_2$ NPs gradually shifts to a neutral dominant electron density, where the red region, blue region and white region indicate the lower electron density, densely electron-dense region and neutrality, respectively (Fig. 3e). This calculation results indicate that the high crystallinity and abundant C-Si covalent bonds formed under high-temperature calcination (575 °C) can render the molecular structure more stable, which provides a rigid environment for restricting excited molecular motions of luminescent centers, and thus suppressing non-radiative processes[35]. In addition, the HOMO-LUMO energy gap of the CDs decreases from 3.0928 to 2.0237 eV with the increase of the calcination temperature (Fig. 3f and Supplementary Fig. 29). The lower the HOMO-LUMO energy gap, the higher the efficiency in the generation and capture of electron excitons, which facilitates the formation and transition of triplet excitons.

Based on the aforementioned analysis, the internal carbon source within $SiO_2$ NPs and the calcination process are the two critical factors for achieving RTP. Therefore, we deduce that the preparation of spherical RTP $SiO_2$ NPs is irrelevant to the type of introduced carbon-containing organic low-molecular-weight molecules during the Stöber method. As illustrated in Fig. 3g, in addition to glucose, 12 different kinds of organic molecules are selected, including nitrogen-containing organic molecules and fluorescent dye molecules. Evidently, the $SiO_2$ NPs doped with these molecules all exhibit long-lived RTP decay characteristics after calcination (Supplementary Fig. 30), demonstrating the universality of this strategy. The inexpensive of the precursor and convenient preparation method ensures the feasibility of large-scale production of RTP $SiO_2$ NPs. By proportionally scaling up the reaction precursors during the synthesis process, it is possible to effortlessly produce a large quantity of spherical RTP nanospheres, with the mass exceeding 700 g (Fig. 3h, Supplementary Fig. 31 and Supplementary Table 4).

## PBG-induced angle-dependent chromatic behavior of PC gels

Subsequently, RTP $SiO_2$ NPs with different diameters are dispersed in a mixture of ethanol, trimethylolpropane ethoxylate triacrylate (ETPTA), poly(ethylene glycol) diacrylate (PEGDA), and ethylene glycol (EG) to prepare blue, green and red photonic gels (B-, G-, R-PC gels) using a supersaturated evaporation-induced self-assembly method in solvent. These PC gels possess SC, FL and RTP simultaneously under different light stimulation conditions (Fig. 4a). However, there are significant differences in the emission intensities and color of FL and RTP for the three PC gels. Evidently, the PBG generated by the ordered array of RTP $SiO_2$ NPs through non-close packing exerts a regulatory influence on their own FL and RTP emissions (Fig. 4b, Supplementary Fig. 32). When the emission band of FL and RTP align with the PBG, it means the transition energy levels of the molecules match the lattice constant of the photonic crystal, the excited photons localize in the micro-nano structure of the PCs, enhancing the resonance efficiency between the emitted light and the photonic structure (Fig. 4c). As a result, the photon radiation recombination rate increases, thereby enhancing the emission intensity (Supplementary Fig. 33)[36–39]. Therefore, the FL emission is the strongest in the B-PC gel, whereas the RTP exhibits the strongest emission in the G-PC gel.

On the other hand, owing to the broad emission characteristics of FL and RTP in PC gel, it is possible to realize PBG-induced color variations in FL and RTP through the modulation of the PBG to facilitate resonance enhancement of a specific emission bandgap. As depicted in Fig. 4d, the angle-dependent multimodal photonic device is fabricated by spin-coating R-PC gel with a thickness of 0.5 mm onto a black light-absorbing substrate and covering it with high-transparency quartz glass. Then, the incident light area (white or UV light) and the light-signal reception area (SC, FL, and RTP) are partially separated using a black opaque light-absorbing plate. Fixing the incident light angle ($\theta_1$) at 90°, the angle-dependent PBG gradually blue-shifts from 611 to 405 nm as the observation angle ($\theta_2$) decreases from 90° to 30° (Supplementary Fig. 34). As anticipated, the blue shift of the PBG leads to a noticeable shift in the additional PBG resonance peaks of the FL and RTP emissions from the R-PC gel (Fig. 4e, f). When the emission wavelengths of FL and RTP fall within the PBG range of PC gel, their propagation is significantly suppressed. These specific wavelength bands of FL and RTP undergo multiple reflections within the PC structure, which enhances photon-matter interaction efficiency and remarkably improves the resonant coupling efficiency between luminescent molecules (or excitons) and the PBG of photonic crystals. Furthermore, due to the anisotropic property of the PC gel structure, the PBG exhibits direction-dependent bandgap characteristics in different spatial orientations, which enables specific FL and RTP wavelength bands to satisfy matching conditions only within predetermined angle ranges, thereby inducing directional resonant coupling. Therefore, the overall detected FL and RTP spectra show a superposition of the constant FL or RTP peaks with the angle-dependent PBG resonance enhancement peaks, resulting in an angle-dependent chromatic behavior. Interestingly, as the observation angle decreases from 90° to 30°, the color changes of three optical signals in the R-PC gel are visible to the naked eye. Under white light illumination, the SC of the R-PC gel gradually changes from red to violet as the observation angle decreases from 90° to 30°, and under UV light illumination or turned off, the FL of the R-PC gel gradually transitions from pink to blue, while the RTP shifts from yellow-green to cyan (Fig. 4g and Supplementary Movie 3). It is worth mentioning that the angle-dependent chromatic behavior for FL and RTP only exists in highly ordered photonic crystal structures (with angle-dependent PBG), while it is difficult to observe such chromatic behavior for poorly ordered PC structures (Supplementary Fig. 35). Besides, as the observation angle changes, the PBG gradually matches with the emission wavelengths, enabling the detection of stronger FL and RTP signals, and the enhancement effects of FL and RTP are most pronounced at $\theta_2 = 40°$ and $\theta_2 = 50°$, respectively (Fig. 4h and Supplementary Fig. 36). The above angle-dependent chromatic behavior of FL and RTP should also exist in the B-PC gel and G-PC gel (Supplementary Figs. 37–39).

The influence of the incident angle of excitation light on the optical behavior of PC gel is further investigated. As the incident angle ($\theta_1$) of the UV light decreases from 90° to 30°, the wavelengths of the FL and RTP spectra measured at $\theta_2 = 90°$ remain constant, while the emission intensities gradually decrease (Fig. 4i and Supplementary Fig. 40). Since the PBG of PC gel is located in the violet band at a low angle, the propagation of external incident UV light is prohibited when it is incident on the PC gel surface at a low incidence angle, and the UV light is reflected from the structure by Bragg reflection, which reduces the excitation energy delivered to the embedded RTP $SiO_2$ NPs, thereby suppressing the emission intensity of FL and RTP[40]. Meanwhile, the incident angle of the excitation light also affects the RTP decay lifetime of the PC gel (Fig. 4j and Supplementary Fig. 41). Based on the above discussion, the FL and RTP of the PC gels are affected by both the incident angle of excitation light and the observation angle, where the former is mainly responsible for the modulation of the emission intensity, while the latter is used to realize the angle-dependent chromatic behavior (Fig. 4k). In short, compared with complex chemical regulation, this angle-dependent physical modulation exhibits transient, highly stable and reproducible characteristics.

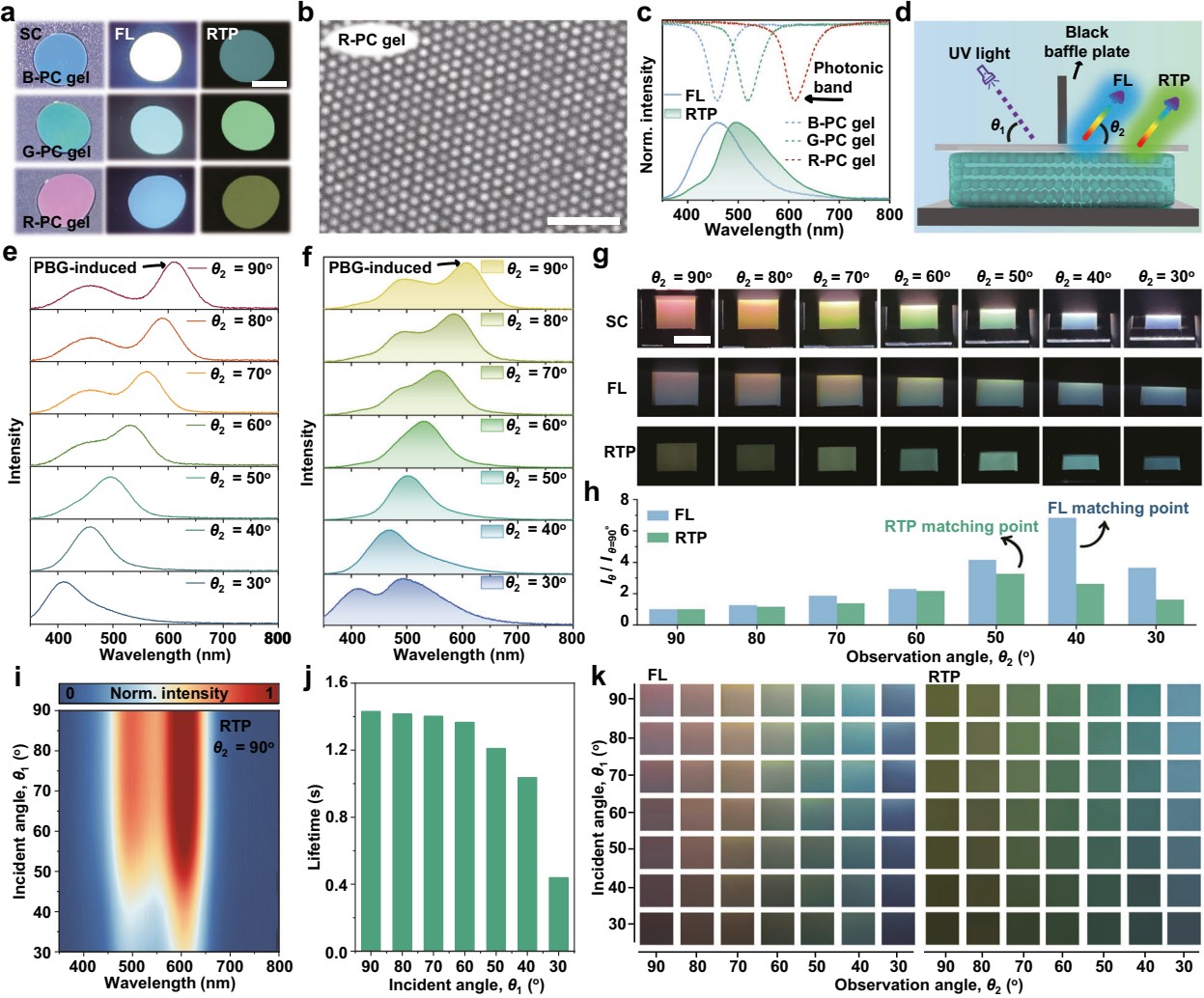

**Fig. 4 | Angle-dependent chromatic behavior of photonic gels. a** Photographs of PC gels with varying particle sizes of RTP $SiO_2$ NPs under daylight, 365 nm UV lamp on and off, respectively, scale bar: 1 cm. **b** The SEM image of R-PC gel prepared by 284 nm diameter of RTP $SiO_2$ NPs (scale bar: 2 μm). **c** The matching situation between the PBG of B-, G-, and R-PC gels with the FL and RTP of RTP $SiO_2$ NPs, the dotted lines, blue solid line, and green fill line represent the photon bandgaps of the three gels, the FL and RTP spectra of RTP $SiO_2$ NPs, respectively. **d** Schematic illustration for achieving angle-dependent chromatic behavior of PC gels. **e**, **f** Angle-dependent FL and RTP spectra of R-PC gel. **g** The photographs of the angle-dependent SC, FL, and RTP of the R-PC gel under daylight, 365 nm UV lamp on and off, respectively, scale bar: 1 cm. **h** The relationship between the intensities of FL and RTP of R-PC gel and the observation angle, where $I_\theta$ is the intensity at different $\theta_2$ and $I_{\theta=90°}$ is the intensity at $\theta_2 = 90°$. **i** Scanning $\theta_1$-dependent two-dimensional RTP spectra of R-PC gel. **j** RTP decay lifetime measured at different $\theta_1$. **k** The color arrays formed by photographs of R-PC gel collected at different $\theta_1$ and $\theta_2$, the size of each unit module is $1 \times 1$ cm². Source data are provided as a Source Data file.

## Self-scattering enhanced luminescence properties of PC gels

The prepared PC gels also have thermal-induced self-scattering enhanced luminescence properties. As shown in Fig. 5a, since R-PC gel is composed of polymer matrix with a temperature-sensitive refractive index (RI) and $SiO_2$ NPs with temperature-independent RI, the gel has temperature-regulated dynamic RI, and the RI-matching point is ≈40 °C. As the temperature decreases from 40 to 0 °C, the RI between the two phases gradually mismatches, resulting in an increase in the light scattering capacity of the R-PC gel, thereby reducing the overall transmittance (Supplementary Fig. 42)[41]. Meanwhile, the strong light scattering at low temperature can also affect the FL and RTP emission. Compared with the RI-matching state at 40 °C, the emission intensity of FL and RTP at 0 °C increase by 128-fold and 87-fold, respectively (Fig. 5b). Besides, the RTP lifetime in the scattered state is extended by 25-fold compared to the transparent state (Fig. 5c). Importantly, the temperature-dependent multiple optical signals of PC gel can be easily captured. As the temperature decreases gradually from 40 to 0 °C, the PC gel turns from transparent state to white scattering state, and the

blue FL is also accompanied by a significant enhancement in this process (Fig. 5d and Supplementary Movie 4). Moreover, in the RI-matched transmittance state, the RTP signal of the PC gel is difficult to capture, while in the scattering state, the RTP signal exhibits a bright and persistent afterglow (Fig. 5e and Supplementary Movie 5). Therefore, the overall optical signals of the PC gel, including transmitted light, scattered light, FL, and RTP, can be modulated by simply changing temperature, and this modulation has a stable reproducibility (Fig. 5f).

The intrinsic physical mechanism of the thermal-induced self-scattering enhanced FL and RTP properties of the R-PC gel is further investigated. The scattering intensity of UV-excited light collected on the PC gel increases with decreasing temperature, suggesting that the scattering state at low temperatures can amplify the intensity of excitation light, thus leading to stronger photoexcitation of the inside RTP $SiO_2$ NPs (Supplementary Fig. 43). In other words, the interfacial scattering enhancement firstly induces multiple omni-directional scattering of the incident UV light inside the gel, which improves the overall

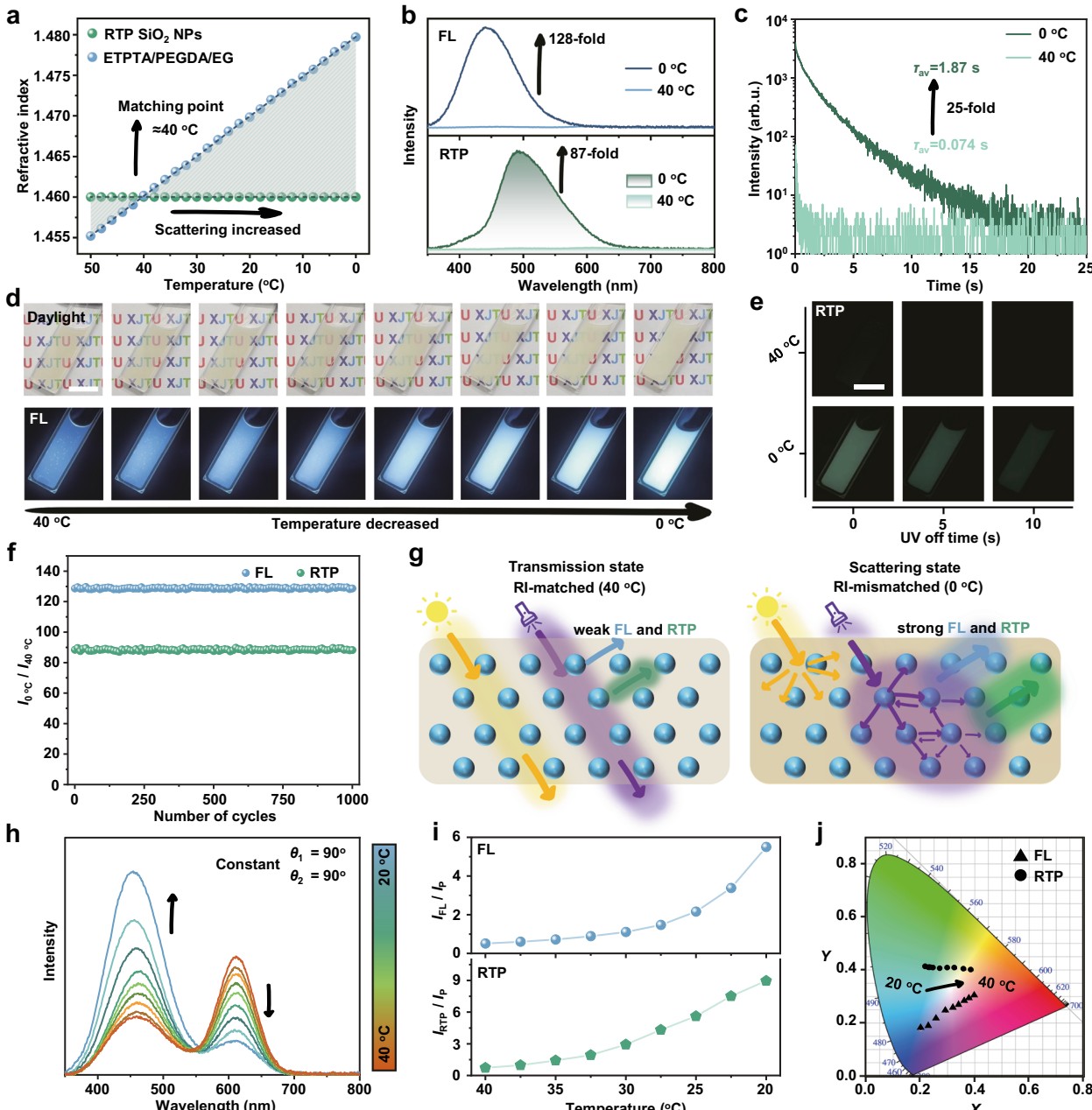

**Fig. 5 | Thermal-induced self-scattering enhanced luminescence and thermo-chromic properties of PC gels. a** Temperature dependence of the refractive index of RTP SiO$_2$ NPs and ETPTA/PEGDA/EG matrix. **b** The FL spectra and RTP spectra of R-PC gel at 0 and 40 ℃, respectively. **c** The RTP lifetime decay spectra of R-PC gel at 0 and 40 ℃, respectively. **d** Photographs of the transparent state of R-PC gel varying with the temperature, and FL photographs of R-PC gel in different temperatures under 365 nm UV light, scale bar: 1 cm. **e** The time-dependent RTP photographs of R-PC gel taken at 0 and 40 ℃, respectively, scale bar: 1 cm. **f** Cyclic performance of

the thermal-induced self-scattering enhanced luminescence properties of R-PC gel. **g** A plausible mechanism for thermal-induced self-scattering enhancement of the FL and RTP. **h** Temperature-dependent FL spectra of R-PC gel, $\theta_1 = 90°$ and $\theta_2 = 90°$. **i** The corresponding relationship between the intensity ratio of self-emission peak ($I_{FL}$ or $I_{RTP}$) and PBG resonance peak ($I_P$) with temperature. **j** The corresponding CIE chromaticity diagram of FL and RTP spectra under different temperatures. Source data are provided as a Source Data file.

excitation efficiency of RTP SiO$_2$ NPs (Fig. 5g)[42]. Although the direct absorption of incident UV light by RTP SiO$_2$ NPs may slightly decrease due to surface scattering enhancement, more scattered UV light will be "captured" inside the gel and interact repeatedly with the particles. This behavior causes electrons to be continuously refilled into the excited state, forming a cycle of excitation, decay and re-excitation, which is equivalent to prolonging the time of electrons during the decay process by sustaining repeated energy transitions, leading to an increase in FL lifetime from 7.7 ns in the transparent state to 11.8 ns in the scattering

state (Supplementary Figs. 44, 45). On the other hand, considering that low temperature limits the thermal motion of molecules in RTP SiO$_2$ NPs, more energy is released through radiative transitions. Therefore, we use a photo-initiator (2-hydroxy-2-methylpropiophenone) to cross-link the gel, making its optical scattering properties independent of temperature (Supplementary Fig. 46), to investigate whether the fundamental mechanism of luminescence enhancement in R-PC gel during cooling from 40 to 0 ℃ originates from scattering enhancement caused by refractive index mismatching. Upon cooling from 40 to 0 ℃, the

crosslinked PC gel exhibits 1.07-fold and 1.13-fold enhancements in FL and RTP intensities, respectively, accompanied by a 1.04-fold prolongation of RTP lifetime (Supplementary Fig. 47). Therefore, the weak enhancement originating from suppressed molecular thermal motion at low temperatures strongly suggests that the refractive index mismatching in gel systems is the main reason for self-scattering-enhanced luminescence. Furthermore, the thermally-induced self-scattering enhancement behavior is independent of the PC structure, relying mainly on the amplification of the excitation light signal in the scattering state (Supplementary Fig. 48).

In addition, the R-PC gel shows enhanced luminescence accompanied by color changes in the temperature interval from 20 to 40 °C. As shown in Fig. 5h, the intensity assigned to the FL peak of R-PC gel at approximately 450 nm gradually increases, while the FL band enhanced by PBG resonance peak at 611 nm gradually decreases with the temperature decreasing from 40 to 20 °C. A similar trend is also observed in the temperature-dependent RTP spectra (Supplementary Fig. 49). Therefore, the thermochromic behavior of R-PC gel is achieved by appropriately modulating the ambient temperature to cause the two fluorescent emissions to compete with each other (Fig. 5i, j). The mismatched refractive index at low temperatures leads to an increase in incoherent light scattering of the R-PC gel, reducing the wavelength selectivity and directionality regulation of light by PBG. Therefore, the relative reflectivity of the gel to a specific red light wavelength band decreases at low temperatures (Supplementary Fig. 50). Meanwhile, this random scattering hinders the directional resonant enhancement interaction of PC gel on FL or RTP through PBG. In conclusion, this R-PC gel composed of spherical phosphor nanoparticles not only integrates multiple optical signals, but also has multi-modal and multi-dimensional stimulus response properties, which provides a potential road for designing intelligent optical devices.

## Discussion

In summary, we demonstrate a scalable approach for synthesizing RTP SiO$_2$ NPs with uniform spherical geometry and low dispersity in size. During high-temperature calcination, the introduced organic low-molecular-weight molecules embedded in silica networks undergo in situ carbonization and crystallization to form CDs, while the rigid environment provided by the C-Si covalent bond efficiently suppresses non-radiative transitions and facilitates the stabilization of triplet excitons, enabling robust RTP emission. The resulting RTP SiO$_2$ NPs retain self-assembly capabilities, generating photonic crystals that synergistically integrate structural color, FL, and RTP into a single multimodal optical platform. To harness dynamic interactions between PBG effects and FL or RTP emission, we engineered metastable multimode PC gels with angle-dependent PBG modulation. These gels exhibit angle-chromatic responses, where light propagation is modulated by the angle-dependence of PBG, resulting in rare FL-RTP angle responses. Furthermore, the thermally tunable refractive index mismatch between SiO$_2$ NPs and the liquid matrix enables dynamic modulation of light scattering and transmission pathways, yielding temperature-gated self-scattering effects that amplify both FL and RTP intensities. The strategy of large-scale preparation of RTP SiO$_2$ NPs with a low dispersity in size and the successful integration of physical photonic structure and photoluminescence provide an alternative approach for constructing advanced multimode luminescent devices.

## Methods
### Materials
Ammonia (28%), trimethylolpropane ethoxylate triacrylate (average $M_n \approx 428$ g mol$^{-1}$), poly(ethylene glycol) diacrylate (PEGDA average $M_n \approx 400$ g mol$^{-1}$), ethylene glycol (EG, 98%), glucose (98%), tartaric acid (99%), $p$-phthalic acid (99%), vitamin C (99%), urea (99%), dopamine (98%), melamine (99%), thiourea (99%), rhodamine B (98%), rhodamine

6 G (95%), quinine sulfate (98%), and sulforhodamine B (95%) are obtained from Shanghai Aladdin Reagent Co., Ltd. Tetraethyl orthosilicate (98%) is purchased from Damao Chemical Reagent Factory. Ethanol (99.7%) is purchased from Tianjin Fuyu Fine Chemical Co., Ltd. All reagents are of analytical grade and used directly without further purification. Deionized water is produced through a Millipore water purification system (Milli-Q, Millipore) and used throughout the study.

### Synthesis of RTP SiO$_2$ NPs
RTP SiO$_2$ NPs is synthesized through a modified Stöber method and their sizes can be regulated by adjusting ethanol ratio during the preparation process. Firstly, 6 mL of glucose aqueous solution (1 mol L$^{-1}$), 130 mL of ethanol and 16 mL of ammonia are mixed homogeneously in a 500 mL three-necked flask and heated to 60 °C. Then, the mixture of 12 mL of tetraethyl orthosilicate and 10 mL ethanol is added and stirred at 60 °C for 100 min in a water bath environment. The white suspension obtained is repeatedly centrifuged three times with ethanol at 5203 × g. The collected white precipitate is dried at 50 °C for 48 h to obtain organic small molecule doped SiO$_2$ NPs. Then, quartz crucible with organic small molecule doped SiO$_2$ NPs is transferred to a high temperature box-type electric resistance furnace, heated up to 575 °C in 3 h and kept for 2 h, finally cooled naturally to room temperature and obtained the RTP SiO$_2$ NPs.

### Preparation of PCs assembled by RTP SiO$_2$ NPs
The RTP SiO$_2$ NPs are dispersed in ethanol to form a white suspension (4 wt.%). Then, the commercial glass is stabilized in the suspension at an inclination angle of 30°. The ethanol is evaporated at an ambient temperature of 50 °C to allow the RTP SiO$_2$ NPs to self-assemble on the glass sheet to form PCs. The PCs with different colors can be prepared by selecting different particle sizes of RTP SiO$_2$ NPs.

### Preparation of PC gels
Briefly, RTP SiO$_2$ NPs (0.12 cm$^3$) are dispersed in a mixture of ethanol (3 mL), ETPTA (140 μL), PEGDA (70 μL), and EG (70 μL) under sonication. The PC gel is obtained by evaporating the mixture at 90 °C for 2 h to evaporate the ethanol, and the volume fraction of RTP SiO$_2$ NPs, ETPTA, PEGDA, and EG in the PC gel is 30%, 35%, 17.5% and 17.5%, respectively. PC gel with different colors can be prepared with different particle sizes of RTP SiO$_2$ NPs. Subsequently, the prepared PC gel is filled into a customized black light-absorbing plate and quartz glass interlayer to achieve the optical behavior of angle-dependent chromatic.

### Characterization
Optical reflectance spectra, transmittance spectra, FL spectra, and RTP spectra are captured utilizing a fiber optic spectrometer (PG2000-Pro), and all RTP emission spectra are measured at 10 ms delay time after turning the externally equipped UV lamp off. The UV-vis absorption spectra are measured with a PerkinElmer PE Lambda 950 spectrometer. Temperature-dependent afterglow spectra and RTP lifetime decay profile is collected with the FLS1000 instrument equipped with the OXFORD accessory. The absolute photoluminescence quantum yield (PLQY) is measured based on the Edinburgh FLS1000 fluorescence spectrometer equipped with an integrating sphere accessory, the scanning steps are 0.3 nm and the range is 345–800 nm. The morphologies of PCs and PC gel are analyzed using a Zeiss Sigma 300 field-emission scanning electron microscope. TEM images, HAADF-STEM images and their corresponding EDS mapping images are obtained on a JEM-2100 apparatus operating at 200 kV. HR-TEM images are taken by a 300 KV double spherical aberration-corrected electron microscope, JEM-ARM300F2 (Grand ARM). The FT-IR spectra is acquired on a Nicolet iS50 FTIR spectrometer. The XPS measurements are performed on an Thermo Scientific K-Alpha spectrometer. Digital photographs of the device are captured using an Apple iPhone 13.

## Measurement and fitting of RTP and FL lifetime

The phosphorescence lifetime measurements are performed using the dynamic decay mode integrated into the Fluoracle® software of Edinburgh FLS1000 fluorescence spectrometer. Each decay curve was acquired over a total acquisition time of 50 s. Specifically, these samples, including RTP $SiO_2$ NPs powder, self-assembled PCs and PC gels, are subjected to continuous irradiation with a 365 nm xenon lamp for 25 s. Following irradiation cessation, time-resolved RTP decay profiles at 504 nm are recorded during the subsequent 25–50 s interval. The acquired decay curves are analyzed via tri-exponential fitting using the Eq. (2):

$$I(t) = A_1 \exp(-t/\tau_1) + A_2 \exp(-t/\tau_2) + A_3 \exp(-t/\tau_3) \quad (2)$$

where $I(t)$ represents the phosphorescence intensity at time $t$. $A_i$ denotes the pre-exponential factors, and $\tau_i$ corresponds to the decay time. The $\tau_{av}$ is subsequently calculated using the Eq. (3):

$$\tau_{av} = \sum A_i \tau_i^2 / \left( \sum A_i \tau_i \right) \quad (3)$$

Besides, the measurement of FL lifetime is conducted using a 375 nm laser monitored at 466 nm, and analysis protocols for decay curves identical with RTP lifetime.

## Measurement and calculation of PLQY

The PLQY are measured based on the Edinburgh FLS1000 fluorescence spectrometer equipped with an integrating sphere accessory. In detail, the background spectra of the excitation light at 365 nm are scanned first, the scanning steps are 0.3 nm and the range is 345-850 nm. Subsequently, the samples are placed on the sample platform inside the integrating sphere, the mass of the powder samples is guaranteed to be 0.07 g, and the volume of the liquid gel samples is 1 mL. Then, the photoluminescence spectra of these RTP $SiO_2$ NPs are recorded under the same conditions. The calculation of PLQY is accomplished through the quantum yield analysis module of the Fluoracle® software built into the Edinburgh FLS 1000 system. The scattering region of the excitation light ($\approx$ 350-378 nm) and the fluorescence emission region of the sample ($\approx$ 378-780 nm) are respectively set as integral intervals, and the values are automatically calculated in combination with the absolute quantum yield Eq. (4):

$$PLQY = \frac{\text{Emitted photons}}{\text{Absorbed photons}} = \frac{\int L_{sample}(d\lambda)d\lambda}{\int \left[ E_{ref}(d\lambda) - E_{sample}(d\lambda) \right] d\lambda} \quad (4)$$

Where $L_{sample}$ represents the emission spectra of the sample, and $E_{ref}$ and $E_{sample}$ are the excitation luminous fluxes without and with the sample, respectively.

## Theoretical calculations

Quantum chemical studies are performed employing density functional theory (DFT) implemented in the Optimal Reciprocal Collision Avoidance (ORCA) program. To explore the mechanism by which stiff C-Si bonds, dense silica-oxygen structure and the degree of crystallinity of CDs affect the phosphorescence characteristics of CDs. Based on the results of, HR-TEM, FT-IR spectroscopy, and XPS characterization, there are variations in the Si-O-C and Si-C covalent bonds from CDs-325 °C, CDs-475 °C, and CDs-575 °C, which are chosen as the study models of CDs-325 °C, CDs-475 °C, and CDs-575 °C, respectively. The input files are generated employing Avogadro. Geometry optimization of CDs-325 °C, CDs-475 °C and CDs-575 °C models is initially performed at the B97-3c level, followed by single-point-energy (SPE) calculations employing the high-precision basis set B3LYP/def2-TZVP. To gain a more thorough understanding of the electronic structure, it is further investigated that SPE calculations provide the basis for the distribution of frontier molecular orbitals and the electrostatic potential (ESP). The molecular visualization programs Jmol and Chemcraft are used for the analysis.

## Data availability

The data that support the findings of this study are available from the corresponding authors upon request. Source data are provided with this paper.

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

## Acknowledgements

We acknowledge funding of the National Key R&D Program of China (2022YFE0109500, S.Y.), the National Natural Science Foundation of China (62475212 and 61975162, X.L.), the Space Application System of China Manned Space Program (No. KJZ-YY-NCL10, S.Y.), and the Shaanxi Province Natural Science Foundation (No. 2020JM-062, X.L.). We thank Dr. D. He at the Instrument Analysis Center of Xi'an Jiaotong University for the kinetic decay curves spectra experiments. We thank the Xi'an Yizhichen Biotechnology Co., Ltd. for their help in purchasing reagents for this experiment.

## Author contributions

C.W. and X.L. conceived the concept and idea. The overall experimental design was completed by C.W., Y.N. and Y.Y. and together, they coordinated the full project. C.W. and Y.N. designed the materials and conducted experiments. C.W., X.W., Y.X. and J.L. characterized, and analyzed the materials. C.W., G.D. and J.L. built angle-dependent chromatic gel equipment and performed data collection and analysis. All authors also discussed and interpreted the experimental data and results. C.W. and N.B. performed the theoretical calculations. C.W. wrote the manuscript with contributions from all authors. X.L. and S.Y. supervised the project and reviewed the paper. All authors discussed the results and commented on the paper.

## Competing interests

The authors declare no competing interests.
