## [Transparent Peer Review file · Nature Communications]

Scalable synthesis of phosphorescent SiO₂ nanospheres and their use for angle-dependent and thermoresponsive photonic gels with multimode luminescence

Corresponding Author: Xuegang Lu

Version 0:

Reviewer comments:

Reviewer #1

(Remarks to the Author)

The work of Wang et al describes a method for fabricating monodisperse phosphorescent nanospheres with a modified Stöber synthesis. The particles are shown to be scalable as the authors have made a batch with more than 700 g. The luminescent silica with impurities was shown already a while ago (10.1021/la703392m, 10.1021/cm060664t), others have also shown it for titania with a similar approach (<https://pubs.acs.org/doi/10.1021/la103717m>). Authors have successfully reproduced fluorescence with glucose but also fabricated similar observations with other impurities. They also report phosphorescence (PS) with the same impurities, which was not shown in the early work of Schmedake et al. (10.1021/cm060664t). They also explored the origin of the fluorescence (FL) by dissolving the silica matrix and looking at the remaining carbon impurities at varying temperatures. They found that the luminescence and phosphorescence increase with T, and it correlates with the growing size of the carbon dots observed as a result of sintering carbon impurities. Authors make sound claims about the origin of the luminescence and phosphorescence and their relation to the increasing Si-C bonds observed in the particles as the sintering temperature is increased. I am convinced by the claims and the supporting set of detailed experiments here. In addition to FL and phosphorescence, authors also demonstrate structural color (SC) formation due to the assembly of the silica particles. Combination of SC with luminescent was shown in early works of Aizenberg and co-workers (0.1039/C3TC30919C) but with limited success.

Authors have performed experiments on the angle dependence of the SC and its influence on FL and PS, resulting in angle-dependent variations in FL and PS. This is all very well and the findings are interesting but does not bring an advance for a technological use. All SC, FL and PS may be used alone and this is interesting but authors may want to clarify the advance of having them together and the added value of having them together.

Another point is the line here "The enhancement effects of FL and RTP are most pronounced at $\theta_2=40^\circ$ and $\theta_2=50^\circ$, respectively (Fig. 4h)." This is discussed but the spectra is given with arbitrary units do not demonstrate the enhancement but rather the overlap of the peaks. Authors may reproduce it in SI to demonstrate the enhancement.

Authors discuss in line 140 that "In addition, RTP SiO₂ NPs can also achieve PC array in the form of 140 simple cubic close-packed arrangement (Supplementary Fig. 9)." They have to be careful calling this because simple cubic is not favorable for spheres and they may simply observe the 110 or 100 face of the fcc crystal. The SI Fig 9 looks to me 100 plane of fcc. In simple cubic, layers should stack in AAA fashion; however, in SI Fig 9 it looks as it is alternating. Therefore, I believe this is fcc but only growing on another face, which is observed for fcc. See this paper for more <https://www.nature.com/articles/385321a0>

Line 396 " The strong scattering at low temperatures induces a decrease in the reflectivity of the PC gel, hindering the resonance-enhanced interactions of the PBG for specific bands of FL or RTP (Supplementary Fig. 42). "

This line was not clear to me , strong scattering is expected to increase reflection in my view. The authors may want to clarify

this or correct it if there is a typo.

Line 319 FL and RTP also exists-- should be exists.

Line 273 there seems a typo here --trimethylolpropane triarchy-late (ETPTA),

Line 195 transitions of electrons is weaken ---weakened

Line 151 reflect light----reflected light

Reviewer #2

(Remarks to the Author)

Wang used the improved Stöber method to prepare RTP SiO₂ nanospheres in large quantities through the in-situ calcination of organic molecules, followed by the synthesis of RTP SiO₂-based photonic crystals and photonic gel by evaporation-induced self-assembly. The resulting photonic gel exhibit angle-dependent multimodal modulation of structure color, fluorescence and phosphorescence, as well as temperature-dependent chromatic behavior arising from refractive index changes associated with temperature-induced self-scattering effects.

However, the modulation of structural color due to self-scattering in RTP SiO₂ PCs, as well as the temperature-responsive behavior of fluorescence and phosphorescence, has already been reported. Therefore, I believe this work lacks sufficient novelty. Furthermore, the following aspects need improvement:

1. Line 130: "In addition, the optical intensity of RTP SiO₂ NPs also maintains excellent stability even in different organic solvents and metal ion solutions (Supplementary Fig. 8)".

-This statement lacks corresponding phosphorescence data to support the claim of stability. From photographs alone, it is not possible to determine whether the phosphorescence lifetime remains stable. Notably, the group containing Fe³⁺ appears to show phosphorescence quenching, which should be discussed and explained.

2. Line 143: "As shown in Fig. 2i, the assembled multimodal PC displays angle-dependent structural colors, bright blue FL and time-dependent green RTP under different light stimulation (Supplementary Movie 2).", – To validate the claim of "time-dependent green RTP," time-resolved spectra should be provided, since the current visual evidence only shows that the phosphorescence color remains green over time, without proving a dynamic temporal change.

3. Supplementary Figure 36: It would be clearer to include the fitted lifetime values alongside the decay curves to make the data more intuitive and quantitative.

4. "As shown in Fig. 5a, since R-PC gel is composed of polymer matrix with a temperature-sensitive refractive index (RI) and a temperature-independent SiO₂ NPs"

– Please verify this claim, as Supplementary Fig. 7 indicates a slight redshift in the maximum emission peak of SiO₂ NPs at elevated temperatures, suggesting some temperature dependence in the optical properties of the particles.

5. "As the temperature decreases from 40°C to 0°C, the refractive index between the two phases gradually mismatches, resulting in an increase in the light scattering capacity of the R-PC gel, thereby reducing the overall transmittance (Supplementary Fig. 37)."

– Since the gel matrix consists of multiple components, how was the influence of possible low-temperature-induced crystallization or phase transitions within the matrix ruled out? For instance, does a gel without SiO₂ NPs show similar temperature-dependent transmission changes?

6. "Meanwhile, the strong light scattering at low temperature can also affect the FL and RTP emission. Compared with the RI-matching state at 40°C, the emission intensity of FL and RTP at 0°C increase by 128-fold and 87-fold, respectively (Fig. 5b). Besides, the RTP lifetime in the scattered state is extended by 25-fold compared to the transparent state (Fig. 5c)."

– It is well known that temperature significantly affects both fluorescence and phosphorescence. At lower temperatures, reduced molecular vibrations suppress non-radiative pathways, often enhancing emission intensity. Therefore, how can it be determined that the luminescence enhancement is primarily due to increased scattering rather than intrinsic temperature effects on the chromophores themselves?

Reviewer #3

(Remarks to the Author)

The authors of this paper present their strategy towards the development of a composite nanomaterial displaying multimodal emission.

A two-step synthesis with mass-production capabilities of monodisperse silica nanospheres (SiO₂ NPs) embedding photoluminescent carbon dots (CDs) formed in-situ is reported.

Using such hybrid material a self-assembled photonic crystal (PC) structure is then obtained.

Their clever approach leads to a nanostructure showing optical properties expressed in three different modes at the same time:

1) a structural color arising from the photonic band-gap (PBG) of the PC;

2) a photoemission due to the fluorescence (PL) stemming from the organic CDs;

3) room-temperature phosphorescence (RTP) enhanced by the CDs tightly bound within the SiO₂ NPs structure.

The authors then proceed to study the RTP origin by comparing the properties of NPs obtained at different calcination temperatures concluding that the strong bonds forming between the C and Si atoms render the structure more stable, thus suppressing non-radiative recombination processes and, consequently, enhancing the RTP emission.

Furthermore, the authors prepare gels incorporating the photoluminescent PC structure and study the interaction between PL, RTP and the PBG of the PC gels.

Finally, the thermochromic properties of the prepared PC gels are studied in view of using the compound for the development of thermoresponsive optical devices.

The manuscript is well written and proposes an interesting approach for the incorporation of different optical materials to obtain a hybrid compound with enhanced photonic properties.

This work novelty is found in the synthetic approach which presents mass-production capabilities and allows to overcome the difficulties in designing and manufacturing multi-modal optical devices sensitive to external stimuli.

In fact, works using NPs to obtain PCs which then induce an enhancement in the optical properties of coupled fluorophores and phosphors have already been reported in the literature (10.1002/adma.202415835, 10.1038/s41377-022-01020-2, 10.1002/ange.201907464, 10.1002/adfm.201804429), even using CDs and SiO₂ as component for the hybrid (10.1016/j.cej.2023.142851, 10.1021/acsnan.4c00272).

This work, to the best of my knowledge, would be the first one in which the interplay between PL and RTP of CDs and PC properties of the SiO₂ NPs structure is shown.

Nonetheless, I have some major remarks regarding this work when considering the claimed interplay between the different optical properties of the hybrid material components.

As discussed in the following points, since the importance of this manuscript arises from such optical interaction, the authors should answer these points to prove the importance and novelty of the developed method.

1) When discussing the PC gels, the authors claim that because of the alignment between PBG and emission bands, specific resonant coupling interactions are activated which lead to an enhancement in the energy transfer efficiency between photons which results in stronger emissions (lines 280-283).

This claim is unclear for several reasons.

Firstly, the author should define what they mean with energy transfer between photons.

Energy transfer processes usually occur between different systems, not just photons, with coupled energy levels through processes such as Förster resonance energy transfer.

Secondly, it is unclear how the authors can claim that a stronger emission is observed.

Although the authors provide evidences in which PL or RTP emission is compared at different observation angles, the increase in emission at certain angles may be due simply to a filter effect brought forth by the angle-dependant PBG.

In general, as discussed in the literature regarding PCs (10.1021/cr200063f), the emission of photoactive species found within the PC structure may be influenced by alterations to the local density of states of the overall system that the PC itself causes.

In order to prove this point, nonetheless, the authors would need to accurately measure the quantum yield (QY) of their PC system and compare it to that measured in the absence of PC.

If an actual amplification of the emission is found an increase in QY is expected.

2) When discussing the thermochromic properties of the PC gels, the authors write that because of (a) the overlap between the scattering band of the gels and their own emission band and (b) of the overlap between the excitation band and the emission band, the RTP SiO₂ NPs emission is then scattered in all directions and then continuously transferred, absorbed and re-emitted among particles.

This, the authors claim, would thus enhance the PL and RTP intensity (lines 382-388).

The supposed explanation is counterintuitive as, unless amplified emission arises, a stronger re-absorption of photons would lead to a poorer QY.

This fact can be easily understood by considering the competing non-radiative decay processes which can easily overtake the overall relaxation mechanism after the emitted light is reabsorbed.

The observed increase in intensity, on the other hand, can be easily explained by the suppression of non-radiative decay pathways which normally occurs by lowering the temperature.

These considerations would also explain the increase in emission lifetime observed at lower temperatures.

The authors should clarify these points for instance by studying the temperature dependency of the emission efficiency of RTP SiO₂ NPs and then comparing it to that of PC gels.

Additionally, in the discussion the authors claim that temperature-gated self-scattering effects amplify the emission intensity (lines 429-432).

In view of the above points, also this claim should be reconsidered unless an amplification of the emission is actually shown.

3) Fundamental to the importance of this work is the mass-production capabilities of the developed synthesis.

The authors claim that more than 700g can be obtained in a single batch.

Nonetheless, the synthetic yield of the synthesis (both at low and large scale) should be reported to put the resulting mass-production capabilities into perspective.

4) In order to prove that the embedding of CDs inside SiO₂ NPs can enhance the RTP thanks to the stabilizing role of C-Si bonds, the authors report that the RTP decay lifetime is the longest when calcination is performed at a temperature of 575°C. This point is quite important as it further proves that the SiO₂ structure can effectively stabilize the CDs triplet state by suppressing non-radiative recombination pathways and leading to an enhanced RTP.

Nonetheless, despite the different signal-to-noise ratio, the RTP decay traces reported in the Supplementary Fig. 17 all proceed parallel to each other in a semi-logarithmic plot.

If, as reported by the authors, the RTP lifetime varies between 2s and 0.1s, the slope of the traces should vary of factor of 20 in such plot and not stay parallel.

The authors do not discuss how the time-traces have been analyzed and the reported lifetimes have been obtained.

The least-squares fitting procedure and the model functions utilized to fit the experimental data should be discussed in detail to clarify this point.

If a multiexponential decay has been used as model function, the lifetime and amplitudes of each single exponential should be reported, at least in the supplementary material.

Additionally, the confidence interval of all parameters obtained from the fits should be reported.

5) No details regarding the QY calculations are provided, although QY values are reported in Supplementary Fig. 19 and used to discuss the role in the SiO₂ structure in the enhancement of the CDs emission.

The authors should discuss in detail the method used to obtain the absolute QYs values in the Methods section or in the supplementary material.

Additionally, by observing Supplementary Fig. 16, it appears that at calcination temperatures higher than 575°C the QY of the system decreases.

Why are QYs of the compound obtained at these calcination temperature not reported?

Since (a) crystallinity of the system increases with calcination temperature even above 575°C (lines 197-202) and (b) the authors claim that the crystallinity helps to enhance the QY (lines 193-196), the authors should discuss what leads to the QY reduction above 575°C.

Finally, the confidence intervals associated to the reported QYs values should be indicated as well.

In addition to the major points discussed above, the following minor points should be addressed as well:

6) In the discussion section (lines 420-423) the authors claim that the covalent Si-C network stabilizes the triplet states through spatial confinement.

The term spatial confinement suggests some sort of quantum confinement effect.

Such quantum effect is not discussed in the work while, as also discussed by the authors, the Si-C network helps in the reduction of the non-radiative processes competing with the radiative recombination leading to RTP.

I suggest the authors to rephrase the parts of the text where the term confinement is used to make the discussion clearer and avoid confusion.

7) At lines 132, 208 and 434 the authors use the term chemiluminescence to discuss the optical properties of their system.

Nonetheless, the emission explored in the manuscript does not originates from chemiluminescent processes.

In such kind of processes the energy leading to the emission is provided by chemical reactions.

Differently, in this work the emission always originates from photoexcitation.

I suggest the authors to rephrase all claims of chemiluminescence.

8) At lines 130-133, the authors claim that the RTP SiO₂ NPs present good anti-quenching ability.

Nonetheless in Supplementary Fig. 8 it results that Fe³⁺ ions are able to quench both RTP and PL of the system.

The authors should discuss this point.

9) It is unclear what the difference between the panels (f) and (g) of Fig. 2 is and how the two TEM images have been obtained.

10) At line 472, regarding the preparation of PC gels, a typo is present. The sentence "to ethanol the alcohol" should be corrected.

Version 1:

Reviewer comments:

Reviewer #1

(Remarks to the Author)

I have read all the responses made to my comments and the changes made to the manuscript. I found the response and the modifications convincing to be accepted.

Reviewer #2

(Remarks to the Author)

The revised manuscript has addressed all my concerns and should be accepted.

Reviewer #3

(Remarks to the Author)

I thank the authors for addressing my questions and comments in the response letter as well as in the reviewed manuscript. All of my concerns have been addressed with replies well supported by additional experimental data. I believe that the quality of the manuscript has been enhanced by the additional discussion and performed experiments. I recommend it for publication as it is.

Re: Manuscript reference **NCOMMS-25-24848**

Manuscript title: Scalable Synthesis of Self-Assembling Monodisperse Phosphorescent Nanospheres Enabling Multi-Mode Angle-Dependent and Thermal-Responsive Photonic Gels

Dear Editor,

We would like to thank the reviewers for critical reading our manuscript and their valuable insights which will improve the quality of our manuscript. With incorporating their constructive suggestions, we believe our manuscript will become better and beneficial to provide synthetic strategy for the large-scale preparation of RTP materials with functionalized and adjustable micro/nano structure, as well as achieving novel optical chromatic behaviors regulated by physical microstructures.

Point-to-point responses to all comments raised by the reviewers are attached here with this letter. The revised manuscript, together with an updated supplementary information, have been resubmitted online. We hope our responses listed below bring more clarifications and make our manuscript suitable for publication in *Nature Communications*.

We would greatly appreciate your time and are looking forward to hearing your favorable decision soon.

Yours sincerely,

Authors

Note: The reviewers' comments are in black, and our Response are highlighted in blue.

Responses to the comments of Reviewer #1

The work of Wang et al describes a method for fabricating monodisperse phosphorescent nanospheres with a modified Stöber synthesis. The particles are shown to be scalable as the authors have made a batch with more than 700 g. The luminescent silica with impurities was shown already a while ago (10.1021/la703392m, 10.1021/cm060664t), others have also shown it for titania with a similar approach (<https://pubs.acs.org/doi/10.1021/la103717m>). Authors have successfully reproduced fluorescence with glucose but also fabricated similar observations with other impurities. They also report phosphorescence (PS) with the same impurities, which was not shown in the early work of Schmedake et al. (10.1021/cm060664t). They also explored the origin of the fluorescence (FL) by dissolving the silica matrix and looking at the remaining carbon impurities at varying temperatures. They found that the luminescence and phosphorescence increase with T, and it correlates with the growing size of the carbon dots observed as a result of sintering carbon impurities. Authors make sound claims about the origin of the luminescence and phosphorescence and their relation to the increasing Si-C bonds observed in the particles as the sintering temperature is increased. I am convinced by the claims and the supporting set of detailed experiments here. In addition to FL and phosphorescence, authors also demonstrate structural color (SC) formation due to the assembly of the silica particles. Combination of SC with luminescent was shown in early works of Aizenberg and co-workers (0.1039/C3TC30919C) but with limited success.

Response: We are grateful to the reviewer for the positive assessment of the quality and the recognition of our work. Our work is dedicated to the scalable fabrication of functionalized RTP material featuring nano-spherical morphology and self-assembly characteristics. These functionalized RTP materials enable the construction of photonic crystal structures that synergistically integrate physically modulated light (e.g., reflected, scattered and transmitted light) with photoluminescence, to achieve synergistic interactions between optical signals, as well as exploring a variety of

interesting optical modulation behaviors. In the revised manuscript, we have made our best efforts to address your concerns and added the point-to-point answers to all comments raised by you in this letter.

1. Authors have performed experiments on the angle dependence of the SC and its influence on FL and PS, resulting in angle-dependent variations in FL and PS. This is all very well and the findings are interesting but does not bring an advance for a technological use. All SC, FL and PS may be used alone and this is interesting but authors may want to clarify the advance of having them together and the added value of having them together.

Response: Thank you for your comments. In the manuscript, the realization of the stimulus-responsive chromatic of RTP through the synergistic interaction between different optical signals is the focus and innovation of our research. Different from traditional RTP chromatic regulation methods (such as photochromism, energy transfer, and atomic doping, etc.), the regulation strategy we designed focuses on the physical structure modulation, which not only avoids destroying the inherent optical state of the RTP material, but also makes the regulation more stable and flexible. In the manuscript, we have demonstrated two optical regulation methods. One is to adjust the propagation states of FL and RTP based on the PBG generated by the periodic microstructure, and the other is to enhance FL and RTP by regulating the light scattering ability of the gel through changing temperature. Obviously, these two interesting optical behaviors have great application potential in intelligent sensing, optical anti-counterfeiting and display devices. However, the core value of our research is to break through the traditional RTP regulation ideas and establishing a new type of regulation method of "physical structure regulation-optical signal synergy", thus specific practical applications are not involved in our research. In the manuscript, we focus more on demonstrating the preparation and regulation methods of RTP materials, covering the design and preparation of micro-nano morphological structures, detailed formation mechanisms of RTP, and optical modulation of physical structures. We believe these aspects held more fundamental significance compared to specific applications. Of course, in future works, we will try

to demonstrate the practical applications of such materials in various scenarios.

- Another point is the line here "The enhancement effects of FL and RTP are most pronounced at $\theta_2=40^\circ$ and $\theta_2=50^\circ$, respectively (Fig. 4h)." This is discussed but the spectra is given with arbitrary units do not demonstrate the enhancement but rather the overlap of the peaks. Authors may reproduce it in SI to demonstrate the enhancement.

Response: Thanks for the reviewer's suggestion. As suggestion, we have added the unnormalized spectra (Fig. R1) to the Supplementary Information file (Supplementary Fig. 36) and made corresponding modifications The revision is highlighted in red in the manuscript and listed below:

On page 16: *"and the enhancement effects of FL and RTP are most pronounced at $\theta_2=40^\circ$ and $\theta_2=50^\circ$, respectively (Fig. 4h and Supplementary Fig. 36)."*

Fig. R1: a and b Angle-dependent FL and RTP spectra of R-PC gel. c and d Angle-dependent FL and RTP spectra of G-PC gel. e and f Angle-dependent FL and RTP

spectra of B-PC gel.

3. Authors discuss in line 140 that "In addition, RTP SiO₂ NPs can also achieve PC array in the form of simple cubic close-packed arrangement (Supplementary Fig. 9)." They have to be careful calling this because simple cubic is not favorable for spheres and they may simply observe the 110 or 100 face of the fcc crystal. The SI Fig 9 looks to me 100 plane of fcc. In simple cubic, layers should stack in AAA fashion; however, in SI Fig 9 it looks as it is alternating. Therefore, I believe this is fcc but only growing on another face, which is observed for fcc. See this paper for more <https://www.nature.com/articles/385321a0>

Response: We appreciate your professional comments. To characterize the assembly behavior of RTP SiO₂ NPs, TEM is employed to visually resolve their packing structures. Fig. R2a and b (Fig 2f and g in manuscript) present the RTP SiO₂ NPs in a double-layer and triple-layer close-packed structure, respectively. In Fig. R2a, the double-layer of RTP SiO₂ NPs is clearly observed to nest within the voids formed by the first-layer close-packed spheres, which is a characteristic arrangement with obvious gaps between two layers. By contrast, there are no visible gaps in the three-layer assembly (Fig. R2b), as the NPs in the third layer occupy the gaps created between the first two layers. This results in a stacking sequence where the third layer adopts a distinct stacking form relative to the first two layers, i.e., ABC-stacking, manifesting a typical face-centered cubic (fcc) packing form (Fig. R2c). Therefore, Fig. 2f and Fig. 2g in the manuscript present the close-packed plane of fcc ((111) plane), while Fig. R2d and e (Supplementary Fig. 10 in manuscript) show the (100) plane of fcc.

We have corrected this error in the revised manuscript and the revision is highlighted in red in the manuscript and listed below:

On page 6: *"As presented in Fig. 2f and g, the RTP SiO₂ NPs exhibit an ordered close-packed state with two or more layers, respectively, which is attributed to the (111) plane of the face-centered cubic (fcc). Notably, we also observed the characteristic arrangement pattern of the (100) planes of the fcc structure (Supplementary Fig. 10)."*

On page 9 (Figure legend): *"f and g TEM images of RTP SiO₂ NPs in a double-*

layer and triple-layer close-packed structure, respectively (Scale bar: 500 nm)."

Supplementary Fig. 10 (Figure legend): "a TEM and b SEM images of (100) plane of RTP SiO₂ NPs stacked in fcc structure. (Scale bar: 500 nm)."

Fig. R2: a and b TEM images of RTP SiO₂ NPs in a double-layer and triple-layer close-packed structure, respectively. c Schematic of the stacking form of RTP SiO₂ NPs. d TEM and e SEM images of (100) plane of RTP SiO₂ NPs stacked in fcc structure.

4. Line 396 "The strong scattering at low temperatures induces a decrease in the reflectivity of the PC gel, hindering the resonance-enhanced interactions of the PBG for specific bands of FL or RTP (Supplementary Fig. 42)." This line was not clear to me, strong scattering is expected to increase reflection in my view. The authors may want to clarify this or correct it if there is a typo.

Response: We appreciate the reviewer's insightful comment. The reduced reflectivity we describe in this statement refers to the reduced relative reflectivity of the PC gel in the red light wavelength range (around 611 nm or so). As the temperature decreases from 40 °C to 0 °C, the refractive indices of the two phases (RTP SiO₂ NPs and polymer matrix) that make up the R-PC gel is mis-matched, which leads to an increase in incoherent light scattering at the interface between the two phases (Fig. R3a). Although

the overall scattering intensity will increase (causing the gel to change from transparent to white), this random scattering lacks wavelength selectivity and direction consistency, which disrupts the coherent reflection in the PBG for a specific red light wavelength range, leading to a decrease in relative reflectivity in this range (Fig. R3b). Therefore, the reflection peak of R-PC gel measured at low temperatures at 611 nm is significantly reduced. Meanwhile, the increase of this random scattering hinders the wavelength selectivity and directivity of the PBG enhancement effect, making it impossible for PC gel to precisely regulate the radiation propagation modes of FL and RTP through PBG resonance (Fig. R3c).

In order to improve the presentation of the manuscript, we have revised the inaccurate statement. The revision is highlighted in red in the manuscript and listed below:

On page 20: *"The mis-matched refractive index at low temperatures leads to an increase in incoherent light scattering of the R-PC gel, reducing the wavelength selectivity and directionality regulation of light by PBG. Therefore, the relative reflectivity of the gel to a specific red light wavelength band decreases at low temperatures (Supplementary Fig. 50). Meanwhile, this random scattering hinders the directional resonant enhancement interaction of PC gel on FL or RTP through PBG."*

Supplementary Fig. 50: *"Temperature-dependent calibrated relative reflection spectra of R-PC gel, $\theta_1=90^\circ$ and $\theta_2=90^\circ$."*

Fig. R3: **a** The reflection spectra (absolute reflectance) of R-PC gel without baseline calibration at 0 °C and 40 °C. **b** The baseline calibrated reflection spectra (relative reflectance) of R-PC gel at 0 °C and 40 °C. **c** The FL spectra of R-PC gel at 0 °C and

40 0 °C.

5. Line 319 FL and RTP also exists--should be exists.

Line 273 there seems a typo here --trimethylolpropane triarchy-late (ETPTA),

Line 195 transitions of electrons is weaken ---weakened

Line 151 reflect light----reflected light

Response: Thanks for helping us pointed out these errors. The revision is highlighted in red in the manuscript and listed below:

On page 16: *"The above angle-dependent chromatic behavior of FL and RTP should also exist in the B-PC gel and G-PC gel (Supplementary Fig. 37-39)."*

On page 14: *"Subsequently, RTP SiO₂ NPs with different diameters are dispersed in a mixture of ethanol, trimethylolpropane ethoxylate triacrylate (ETPTA), poly(ethylene glycol) diacrylate (PEGDA) and ethylene glycol (EG) to prepare blue, green and red photonic gels (B-, G-, R-PC gels) using a supersaturated evaporation-induced self-assembly method in solvent."*

On page 23: *"Ammonia, tetraethyl orthosilicate, trimethylolpropane ethoxylate triacrylate (average Mn~428 MW), poly(ethylene glycol) diacrylate (PEGDA average Mn~400 MW), ethylene glycol (EG), glucose, tartaric acid, p-phthalic acid, vitamin C, urea, dopamine, melamine, thiourea, rhodamine B, rhodamine 6G, quinine sulphate and sulforhodamine B are obtained from Shanghai Aladdin Reagent Co., Ltd."*

On page 10: *"It can be inferred that as the calcination temperature increases (from 325 °C to 575 °C), the crystalline structure of the CDs derived from glucose becomes more stable, and the non-radiative transitions of electrons is weakened, leading to a higher overall luminescence efficiency (Supplementary Fig. 19 and Fig. 20 and Table 3).^{31"}*

On page 7: *"where d refers to the distance between the adjacent centers of the RTP SiO₂ NPs, and θ is the angle between the PCs surface and the incident light or reflected light."*

Note: The reviewers' comments are in black, and our Response are highlighted in blue.

Responses to the comments of Reviewer #2

Wang used the improved Stöber method to prepare RTP SiO₂ nanospheres in large quantities through the in-situ calcination of organic molecules, followed by the synthesis of RTP SiO₂-based photonic crystals and photonic gel by evaporation-induced self-assembly. The resulting photonic gel exhibit angle-dependent multimodal modulation of structure color, fluorescence and phosphorescence, as well as temperature-dependent chromatic behavior arising from refractive index changes associated with temperature-induced self-scattering effects.

However, the modulation of structural color due to self-scattering in RTP SiO PCs, as well as the temperature-responsive behavior of fluorescence and phosphorescence, has already been reported. Therefore, I believe this work lacks sufficient novelty. Furthermore, the following aspects need improvement:

Response: We greatly appreciate the reviewer's comments and fully understand why the concern was raised. This work resolves a critical challenge in photonic materials—seamlessly integrating room-temperature phosphorescence (RTP) with dynamically tunable photonic architectures—to create multimodal optical platforms with unprecedented signal control. The core value of our research is to break through the traditional approach of RTP modulation and establish a new type of modulation strategy of "physical structure regulation-optical signal synergy" to surpass the traditional RTP modulation paradigm. Our manuscript provides a systematic solution to three long-standing bottleneck problems in this field: (1) Universal scalable synthesis of monodisperse RTP nanospheres; (2) Dynamic and synergistic modulation of angle-dependent multimodal chromaticity; (3) Synergistic interaction induced self-scattering enhancement of multimodal optical signals, detailed as follows:

Universal scalable synthesis of monodisperse RTP nanospheres: We develop a low-cost, high-yield (>700 g/batch) strategy to synthesize monodisperse RTP SiO₂ nanospheres by embedding organic small molecules (e.g., glucose, Urea) into silica networks, followed by in situ carbonization and crystallization during calcination. This

method eliminates the need for pre-synthesized carbon dots, directly converting small molecules into phosphorescent centers while preserving nanosphere uniformity—bridging the gap between structural precision and RTP functionality. The prepared RTP SiO₂ NPs maintains the monodispersity and self-assembly ability of SiO₂ NPs, while exhibiting a long lifetime of RTP. The periodic array structure formed by self-assembly of RTP SiO₂ NPs not only maintains intrinsic physical-optical properties such as reflection, diffraction, and transmission, but also exhibits high stability and long lifetime RTP in various solvents and most metal ion solutions.

Dynamic and synergistic modulation of angle-dependent multimodal chromaticity: Unlike the currently reported RTP chromatic regulation methods, such as photochromism, thermochromism, energy transfer, etc., we propose a new regulation method for angle-dependent chromatic phenomenon. PC gels assembled from RTP SiO₂ NPs, relying on the PBG generated by their periodic microstructures, can regulate the propagation state of FL and RTP. The PBG of PC gels exhibit orientation-dependent bandgap properties in different spatial orientations, which allows specific FL and RTP wavelengths to satisfy the matching conditions only in a predetermined angle range, causing directional resonant coupling, thus achieving angle-dependent chromatic characteristic. To the best of our knowledge, this is the first example of angle-dependent chromatic of FL and RTP reported so far, and the regulation is dynamic and recyclable.

Synergistic interaction induced self-scattering enhancement of multimodal optical signals: we proposed a novel thermally-induced self-scattering enhanced luminescence behavior, realizing the RTP enhancement behavior of the material through the synergistic interaction between different optical signals. By regulating the refractive index of the two phases of the PC gel from the matched state to the mismatched state through temperature, the PC gel can be switched from the transparent state to the white scattering state, and the increase in light scattering ability greatly enhances the FL and RTP intensities of the PC gel itself. Obviously, compared with traditional enhancement methods, such as surface plasmon resonance, resonance energy transfer and optical microcavities, our method has a high degree of flexibility, reliability and repeatability. Through a systematic literature search, no relevant reports

on the self-scattering luminescence enhancement realized by the synergistic effect of thermally-induced optical signals have been found.

In conclusion, this work establishes a versatile platform for designing stimuli-responsive optical devices, merging nanoscale RTP engineering with microscale photonic control. The scalable synthesis and physical regulation of RTP, as well as thermal-induced self-scattering enhancement of multimodal optical signals, greatly enhance the practical application potential of RTP materials and open up a new avenue for optimizing luminescence performance.

1. Line 130: "In addition, the optical intensity of RTP SiO₂ NPs also maintains excellent stability even in different organic solvents and metal ion solutions (Supplementary Fig. 8)". -This statement lacks corresponding phosphorescence data to support the claim of stability. From photographs alone, it is not possible to determine whether the phosphorescence lifetime remains stable. Notably, the group containing Fe³⁺ appears to show phosphorescence quenching, which should be discussed and explained.

Response: We sincerely thank the reviewer's patient reading and useful comments on our manuscript. In order to comprehensive understanding of the optical stability of the prepared RTP SiO₂ NPs, the FL and RTP intensity for RTP SiO₂ NPs is investigated in the presence of different organic solvents and different kinds of metal ionic interferences with the concentration of 1 M under identical conditions. It is obvious that Fe³⁺ and Co²⁺ can induce FL and RTP quenching for RTP SiO₂ NPs, while other environments or interfering substances show almost no quenching, as demonstrated in Fig. R1a-d. Meanwhile, Fe³⁺ and Co²⁺ also have a significant impact on the RTP lifetime of RTP SiO₂ NPs (Fig. R1e and f). It is worth mentioning that RTP SiO₂ NPs also exhibit a short RTP lifetime in methanol, which is due to low dispersity of RTP SiO₂ NPs in methanol (Fig. R1g). Generally, the quenching of fluorophores by metal ions mainly involves static quenching, dynamic quenching and inner filter effect.¹ Firstly, the FT-IR spectra of RTP SiO₂ NPs in the presence and absence of Fe³⁺ and Co²⁺ show that no obvious shift and no new vibration absorption sites appear (Fig. R2a),

indicating that no functional groups are formed or adhered to the surface of RTP SiO₂ NPs, thus excluding static quenching.² To seek the reason for the quenching phenomenon, the FL lifetime of RTP SiO₂ NPs with and without Fe³⁺ and Co²⁺ are measured. As shown in Fig. R2b, the FL lifetime of RTP SiO₂ NPs in the absence and presence of Fe³⁺ and Co²⁺ are almost unchanged. The inner filter effect typically requires that the absorption band of the quencher overlap with the excitation or emission band of the fluorophore, resulting in a bursting behavior by absorbing the excitation or emission light of the fluorophore.³ As shown in the Fig. R2c and d, the absorption spectra of Fe³⁺ and Co²⁺ ions overlap with the excitation and RTP emission peaks of RTP SiO₂ NPs, respectively, indicating a typical inner filter effect. Furthermore, the RTP SiO₂ NPs after separation from Fe³⁺ and Co²⁺ solutions exhibit reversible recovery of optical properties (Fig. R2e and f).

To make the manuscript more comprehensive and informative, we have supplemented the research results and detailed discussion in the revised Supplementary Information. The revision is highlighted in red in the manuscript and listed below:

On page 6: *"In addition, the optical intensity of RTP SiO₂ exhibits good stability in various organic solvents and most metal ion solutions (Supplementary Fig. 8 and Fig. 9)."*

Supplementary Fig. 9 (Discussion): *"Systematic exploration of the optical stability of RTP SiO₂ NPs in the presence of different organic solvents and different kinds of metal ionic interferences with the concentration of 1 M under identical conditions. Supplementary Fig. 9a-d show the Fe³⁺ and Co²⁺ can induce FL and RTP quenching for RTP SiO₂ NPs. Meanwhile, Fe³⁺ and Co²⁺ also have a significant impact on the RTP lifetime of RTP SiO₂ NPs (Supplementary Fig. 9e and f). It is worth mentioning that RTP SiO₂ NPs also exhibit a short RTP lifetime in methanol, which is due to low dispersity of RTP SiO₂ NPs in methanol (Supplementary Fig. 9g). Generally, the quenching of fluorophores by metal ions mainly involves static quenching, dynamic quenching and inner filter effect.¹ Firstly, the FT-IR spectra of RTP SiO₂ NPs in the presence and absence of Fe³⁺ and Co²⁺ show that no obvious shift and no new vibration absorption sites appear (Supplementary Fig. 9h), indicating that no functional groups*

are formed or adhered to the surface of RTP SiO₂ NPs, thus excluding static quenching.² To seek the reason for the quenching phenomenon, the fluorescence lifetime of RTP SiO₂ NPs with and without Fe³⁺ and Co²⁺ are measured. As shown in Supplementary Fig. 9i, the fluorescence lifetime of RTP SiO₂ NPs in the absence and presence of Fe³⁺ and Co²⁺ are almost unchanged. The inner filter effect typically requires that the absorption band of the quencher overlap with the excitation or emission band of the fluorophore, resulting in a bursting behavior by absorbing the excitation or emission light of the fluorophore.³ As shown in the Supplementary Fig. 9j and k, the absorption spectra of Fe³⁺ and Co²⁺ ions overlap with the excitation and RTP emission peaks of RTP SiO₂ NPs, respectively, indicating a typical inner filter effect."

Fig. R1: FL spectra of RTP SiO₂ NPs dispersed in (a) different solvents and (b) various metal ion solutions. RTP spectra of RTP SiO₂ NPs dispersed in (c) different solvents and (d) various metal ion solutions. RTP decay curve of RTP SiO₂ NPs dispersed in (e) different solvents and (f) various metal ion solutions. g The optical stability of the summarized RTP SiO₂ NPs in different environments, where the I₀ and I represent the emission intensity of RTP SiO₂ NPs dispersed in water and in different environments, respectively.

Fig. R2: **a** FTIR spectra of RTP SiO₂ NPs water dispersion in the presence and absence of Fe³⁺ and Co²⁺. **b** FL decay profile of RTP SiO₂ NPs water dispersion in the presence and absence of Fe³⁺ and Co²⁺. **c** The UV-vis absorption spectra of Fe³⁺, and the FLE and FL spectra of the RTP SiO₂ NPs water dispersion. **d** The UV-vis absorption spectra of Co²⁺, and RTP spectra of the RTP SiO₂ NPs water dispersion. **e** FL and RTP spectra of mixed solutions of RTP SiO₂ NPs with metal ions (Fe³⁺ or Co²⁺) before and after separation. **f** RTP decay curve of mixed solutions of RTP SiO₂ NPs with metal ions (Fe³⁺ or Co²⁺) before and after separation.

References

1. Vervalde, A. M. et al. Quenching of photoluminescence of carbon dots by metal cations in water: Estimation of contributions of different mechanisms. *J. Phys. Chem. C* **44**, 21617-21628 (2023).
2. Wang, C. et al. Synthesis of multi-color fluorine and nitrogen co-doped graphene quantum dots for use in tetracycline detection, colorful solid fluorescent ink, and film. *J. Colloid Interf. Sci.* **602**, 689-698 (2021).

3. Chen, S. et al. Inner filter effect-based fluorescent sensing systems: A review. *Anal. Chim. Acta* **999**, 13-26 (2018).

2. Line 143: "As shown in Fig. 2i, the assembled multimodal PC displays angle-dependent structural colors, bright blue FL and time-dependent green RTP under different light stimulation (Supplementary Movie 2).", –To validate the claim of "time-dependent green RTP," time-resolved spectra should be provided, since the current visual evidence only shows that the phosphorescence color remains green over time, without proving a dynamic temporal change.

Response: Thanks for the reviewer's insightful questions and suggestion. We have measured the RTP spectra of PC self-assembled by RTP SiO₂ NPs (283 nm diameter) collected at different delay times. As shown in Fig. R3a, under the 365 nm excitation, the PC exhibits a characteristic emission peak at approximately 500 nm, which corresponds to the green luminescence region. Further analysis of the chromaticity coordinates reveals that the CIE values at different delay times are all stably distributed around (0.252, 0.381) (Fig. R3b), indicating that the afterglow color of the PCs always exhibits stable green luminescence characteristics.

To make the presentation of manuscript more accurate, we have modified the "time-dependent green RTP" to "long-persistent green RTP".

Fig. R3: **a** Delayed RTP spectra of PC self-assembled by RTP SiO₂ NPs with varying delayed times excited by 365 nm light. **b** CIE coordinate diagram of RTP with varying delayed times of PC.

3. Supplementary Figure 36: It would be clearer to include the fitted lifetime values alongside the decay curves to make the data more intuitive and quantitative.

Response: We thank the reviewer for the nice suggestions. As suggested, we have fitted all the lifetime decay curves in Supplementary Fig. 41 and included the corresponding fitted lifetimes values alongside the decay curves (Fig. R4).

Fig. R4: RTP lifetime decay curves measured at different θ_1 .

4. "As shown in Fig. 5a, since R-PC gel is composed of polymer matrix with a temperature-sensitive refractive index (RI) and a temperature-independent SiO₂ NPs". Please verify this claim, as Supplementary Fig. 7 indicates a slight redshift in the maximum emission peak of SiO₂ NPs at elevated temperatures, suggesting some temperature dependence in the optical properties of the particles.

Response: Thanks to the reviewer for insightful comments. First of all, we apologize for our unclear expression, for Fig 5a in manuscript, we measured the temperature-dependent refractive indices (RI) of SiO₂ and polymers (ETPTA/PEGDA/EG) respectively, and the results show that the RI of SiO₂ is temperature-independent, while the RI of polymers has a significant temperature dependence. Therefore, the PCs gel composed of SiO₂ NPs and polymers has temperature-regulated dynamic refractive

index matching behavior. In this sentence, "temperature-independent SiO₂ NPs", we intend to express that the RI of SiO₂ NPs is temperature independent.

Furthermore, we re-measured the FL and RTP spectra of RTP-SiO₂ NPs at low temperature (77 K) and high temperature (347 K). As shown in Fig. R5a and b, the results of the two measurements both indicate that the FL and RTP of RTP-SiO₂ NPs exhibit temperature responses, and as the temperature increases from 77 K to 374 K, the maximum emission peaks of FL and RTP undergo a red-shift of 12 nm and 43 nm, respectively. Generally, there are two main mechanisms that lead to the temperature-dependent energy gaps: one is the thermal expansion coefficient of the material, and the other is the temperature-related electron-phonon coupling.^{1,2} As the temperature increases, the lattice constant of the material increases due to thermal expansion. The lattice expansion at high temperatures leads to an increase in the quantum size of the CDs inside the RTP-SiO₂ NPs, resulting in a red-shift in the emission wavelength.³ Besides, as the temperature rises, the lattice vibrations are enhanced, and the interaction between electrons and phonons is strengthened, which leads to a red-shift of the emission spectra. Although RTP-SiO₂ NPs have temperature-dependent optical properties, this behavior requires a wide range of temperature variations (77 K-347 K). In thermal-induced self-scattering enhancement behavior, we chose the temperature range of 0 °C-40 °C (273 K-313 K) as the study interval, because this interval not only lies in the dynamic refractive index variation range, but also encompasses ambient temperatures adapted by humans. Within this temperature interval, the emission wavelengths of FL and RTP hardly change with temperature, thus the temperature-dependent optical properties of RTP SiO₂ NPs can be ignored in the thermal-induced self-scattering enhancement behavior (Fig. R5c).

To express accurately, we have revised the inaccurate statement. The revision is highlighted in red in the manuscript and listed below:

On page 18: *"As shown in Fig. 5a, since R-PC gel is composed of polymer matrix with a temperature-sensitive refractive index (RI) and SiO₂ NPs with temperature-independent RI, the gel has temperature-regulated dynamic RI, and the RI-matching point is approximately 40 °C."*

Fig. R5: **a** Raw and **b** Re-measured low-temperature (77 K) and high-temperature (347 K) spectra of FL and RTP of RTP SiO₂ NPs. **c** The FL and RTP spectra of RTP SiO₂ NPs collected at 273 K and 313 K.

References

1. Y, Varshni. Temperature dependence of the energy gap in semiconductors. *Physica* **34**, 149-154 (1967).
2. P, Yu. et al. Temperature-dependent fluorescence in carbon dots. *J. Phys. Chem. C* **116**, 25552-25557 (2012).
3. S, Zhu. et al. Photoluminescence mechanism in graphene quantum dots: Quantum confinement effect and surface/edge state. *Nano Today* **13**, 10-14 (2017).
5. "As the temperature decreases from 40 °C to 0 °C, the refractive index between the two phases gradually mismatches, resulting in an increase in the light scattering capacity of the R-PC gel, thereby reducing the overall transmittance (Supplementary Fig. 37)." Since the gel matrix consists of multiple components, how was the influence of possible low-temperature-induced crystallization or phase transitions within the matrix ruled out? For instance, does a gel without SiO₂ NPs show similar temperature-dependent transmission changes?

Response: We understand the reviewer's concern regarding potential interference from the intrinsic thermally dependent refractive index (RI) behaviors of the three polymeric matrices (ETPTA, PEGDA and EG) themselves on the light scattering modulation in

R-PC gel, and have thoroughly addressed this issue through comparative control experiments. We have measured the temperature-dependent RI of the three polymers (Fig. R6a). The results show that the RI of the three polymers changes at similar rates in the 0-50 °C range and display no crossover points, implying that the gel constructed solely from these polymers lack thermally induced RI matching behavior. Besides, the differential scanning calorimetry (DSC) curves of the three polymers showed no significant crystallization and phase transition characteristic peaks within the temperature range of 0-50 °C (Fig. R6b). Furthermore, the temperature-dependent transmission spectra of gel constructed with three polymers is measured. As shown in Fig. R6c, as the temperature decreases from 50 °C to 0 °C, the gel remains highly transparent state, thus conclusively ruling out interference from intrinsic thermal variations of the polymer matrices on optical modulation. In the manuscript, we constructed R-PC gels by self-assembling RTP SiO₂ NPs within the mixture of three polymers through a supersaturation evaporation method. Due to the distinct temperature-sensitive behaviors of the refractive index between the two phases in R-PC gels, their RI changes at different rates with temperature increasing or decreasing (Fig. R6d). Therefore, the light scattering and transmittance at the interface of the two phases are indirectly regulated by simply modulating the temperature to control the RI matching and mismatching states.

Fig. R6: **a** Temperature dependence of the refractive index of ETPTA, PEGDA and EG. **b** Differential scanning calorimetry (DSC) curves of ETPTA, PEGDA and EG. **c** Temperature-dependent transmission spectra of gel prepared from a mixture of ETPTA, PEGDA and EG. **d** Temperature dependence of the refractive index of RTP SiO₂ NPs and ETPTA/PEGDA/EG matrix.

6. "Meanwhile, the strong light scattering at low temperature can also affect the FL and RTP emission. Compared with the RI-matching state at 40 °C, the emission intensity of FL and RTP at 0 °C increase by 128-fold and 87-fold, respectively (Fig. 5b). Besides, the RTP lifetime in the scattered state is extended by 25-fold compared to the transparent state (Fig. 5c)." It is well known that temperature significantly affects both fluorescence and phosphorescence. At lower temperatures, reduced molecular vibrations suppress non-radiative pathways, often enhancing emission intensity. Therefore, how can it be determined that the luminescence enhancement is primarily due to increased scattering rather than intrinsic temperature effects on the chromophores themselves?

Response: We thank the reviewer for these important questions. In fact, as the reviewer has pointed out, low temperatures can limit the thermal motion of molecules, thereby causing more energy to be released through radiative transitions. To investigate whether

the underlying mechanism of the enhanced luminescence observed in R-PC gel during cooling from 40 °C to 0 °C originates from scattering enhancement induced by refractive index mismatching, the gel is crosslinked using a photo-initiator (2-hydroxy-2-methylpropiophenone) to render their optical scattering properties temperature-independent (Fig. R7a). This experimental design effectively eliminated the interference of temperature-dependent scattering effects on luminescence performance. Upon cooling from 40 °C to 0 °C, the crosslinked PC gels exhibited 1.07-fold and 1.13-fold enhancements in FL and RTP intensities, respectively, accompanied by a 1.04-fold prolongation of RTP lifetime (Fig. R7b, c and d). Notably, these enhancement factors are substantially lower than those observed in non-crosslinked PC gels (128-fold for FL, 87-fold for RTP, and 25-fold for RTP lifetime). The weak enhancement originating from suppressed molecular thermal motion at low temperatures strongly suggests that the refractive index mismatching in gel systems is the main reason for self-scattering-enhanced luminescence. This comparative analysis conclusively validates the feasibility of manipulating luminescence through refractive index-modulated scattering enhancement in photonic crystal materials.

We believe that most readers will question the interference of the inherent temperature effect of fluorophores on the scattering enhancement behavior. To make the manuscript more comprehensive and informative, we have supplemented this research results and detailed discussion in the revised manuscript and Supplementary Information (Supplementary Fig. 46 and 47). The revision is highlighted in red in the manuscript and listed below:

On page 19: *"On the other hand, considering that low temperature limits the thermal motion of molecules in RTP SiO₂ NPs, more energy is released through radiative transitions. Therefore, we use a photo-initiator (2-hydroxy-2-methylpropiophenone) to crosslink the gel, making its optical scattering properties independent of temperature (Supplementary Fig. 46), to investigate whether the fundamental mechanism of luminescence enhancement in R-PC gel during cooling from 40 °C to 0 °C originates from scattering enhancement caused by refractive index mismatching. Upon cooling from 40 °C to 0 °C, the crosslinked PC gel exhibited 1.07-*

fold and 1.13-fold enhancements in FL and RTP intensities, respectively, accompanied by a 1.04-fold prolongation of RTP lifetime (Supplementary Fig. 47). Therefore, the weak enhancement originating from suppressed molecular thermal motion at low temperatures strongly suggests the refractive index mismatching in gel systems is the main reason for self-scattering-enhanced luminescence."

Fig. R7: **a** Transmission spectra of cured R-PC gel with photo-initiators under different temperature. **b** FL, **c** RTP and **d** RTP decay spectra of cured R-PC gel under different temperature.

Note: The reviewers' comments are in black, and our Response are highlighted in blue.

Responses to the comments of Reviewer #3

The authors of this paper present their strategy towards the development of a composite nanomaterial displaying multi-modal emission. A two-step synthesis with mass-production capabilities of monodisperse silica nanospheres (SiO_2 NPs) embedding photoluminescent carbon dots (CDs) formed in-situ is reported. Using such hybrid material a self-assembled photonic crystal (PC) structure is then obtained.

Their clever approach leads to a nanostructure showing optical properties expressed in three different modes at the same time:

- 1) a structural color arising from the photonic band-gap (PBG) of the PC;
- 2) a photoemission due to the fluorescence (PL) stemming from the organic CDs;
- 3) room-temperature phosphorescence (RTP) enhanced by the CDs tightly bound within the SiO_2 NPs structure.

The authors then proceed to study the RTP origin by comparing the properties of NPs obtained at different calcination temperatures concluding that the strong bonds forming between the C and Si atoms render the structure more stable, thus suppressing non-radiative recombination processes and, consequently, enhancing the RTP emission. Furthermore, the authors prepare gels incorporating the photoluminescent PC structure and study the interaction between PL, RTP and the PBG of the PC gels. Finally, the thermochromic properties of the prepared PC gels are studied in view of using the compound for the development of thermoresponsive optical devices.

The manuscript is well written and proposes an interesting approach for the incorporation of different optical materials to obtain an hybrid compound with enhanced photonic properties.

This work novelty is found in the synthetic approach which presents mass-production capabilities and allows to overcome the difficulties in designing and manufacturing multi-modal optical devices sensitive to external stimuli.

In fact, works using NPs to obtain PCs which then induce an enhancement in the optical properties of coupled fluorophores and phosphors have already been reported in the

literature (10.1002/adma.202415835, 10.1038/s41377-022-01020-2, 10.1002/ange.201907464, 10.1002/adfm.201804429), even using CDs and SiO₂ as component for the hybrid (10.1016/j.cej.2023.142851, 10.1021/acsanm.4c00272).

This work, to the best of my knowledge, would be the first one in which the interplay between PL and RTP of CDs and PC properties of the SiO₂ NPs structure is shown.

Nonetheless, I have some major remarks regarding this work when considering the claimed interplay between the different optical properties of the hybrid material components. As discussed in the following points, since the importance of this manuscript arises from such optical interaction, the authors should answer these point to prove the importance and novelty of the developed method.

Response: We are grateful to the reviewer for the positive assessment and the recognition of the quality of our work. Our work is dedicated to the scalable fabrication of functionalized RTP material featuring nano-spherical morphology and self-assembly characteristics. These functionalized RTP materials enable the construction of photonic crystal structures that synergistically integrate physically modulated light (e.g., reflected, scattered and transmitted light) with photoluminescence, to achieve synergistic interactions between optical signals, as well as exploring a variety of interesting optical modulation behaviors. The core value of our research is to break through the traditional approach of RTP regulation and establish a new type of regulation method of "physical structure regulation-optical signal synergy", as commented by the reviewer. In the revised manuscript, we have made our best efforts to address your concerns and added the point-to-point answers to all comments raised by you in this letter.

1. When discussing the PC gels, the authors claim that because of the alignment between PBG and emission bands, specific resonant coupling interactions are activated which lead to an enhancement in the energy transfer efficiency between photons which results in stronger emissions (lines 280-283). This claim is unclear for several reasons. Firstly, the author should define what they mean with energy transfer between photons. Energy transfer processes usually occur between

different systems, not just photons, with coupled energy levels through processes such as Förster resonance energy transfer.

Response: We sincerely thank the reviewer for their thoughtful and professional comments. We apologize for the unclear statement "which enhances the energy transfer efficiency between photons" in our manuscript. Specifically, Förster resonance energy transfer (FRET) typically involves two distinct fluorescent molecules (donor and acceptor), where energy is transferred from the donor molecule to the acceptor. However, in our study, since we use photonic crystal structures to modulate the optical properties of the emitter, and the PC gel contains only one emitter (RTP SiO₂ NPs), the optical modulation is fundamentally an interaction between light and the micro/nanostructure, rather than energy transfer between molecules. Therefore, the use of the term "energy transfer" is inappropriate, and to avoid confusion, we have revised the manuscript by replacing "*which enhances the energy transfer efficiency between photons*" with "*enhancing the resonance efficiency between the emitted light and the photonic structure*".

Additionally, we have provided a more detailed discussion on the emission enhancement mechanism. The FL and RTP spectra for B-, G-, and R-PC gels, as well as poorly ordered PC gel, are measured under identical conditions. As shown in the Fig. R1a and b, FL exhibits the strongest intensity in the B-PC gel, while RTP shows the strongest intensity in the G-PC gel. We measure the FL lifetimes of different gels, and the results show that FL lifetime of the B-PC gel is the shortest (Fig. R1c). This phenomenon is related to the resonant coupling effect of the PCs and the enhancement of the local density of states (LDOS).¹⁻⁴ Specifically, when the emission bands of FL and RTP align with the PBG of the PCs, the excited photons localize in the micro-nano structure of the PCs, enhancing the resonance efficiency between the emitted light and the photonic structure. As a result, the photon radiation recombination rate increases, thereby enhancing the emission intensity. Besides, the strong RTP emission in G-PC gel suggests that the resonance interaction between the emission light and photonic structure is independent of the type of molecular transition, whether S₁-S₀ or T₁-S₀.

Based on the above discussion, we have revised the manuscript to ensure the accuracy and clarity of the manuscript, and have also incorporated the above discussion into the Supplementary Information (Supplementary Fig. 33). The revision is highlighted in red in the manuscript and listed below:

On page 14: *"When the emission band of FL and RTP align with the PBG, it means the transition energy levels of the molecules match the lattice constant of the photonic crystal, and the excited photons are localized in the micro-nano structure of the PCs, enhancing the resonance efficiency between the emitted light and the photonic structure (Fig. 4c). As a result, the photon radiation recombination rate increases, thereby enhancing the emission intensity (Supplementary Fig. 33).³⁶⁻³⁹ Therefore, the FL emission is the strongest in the B-PC gel, whereas the RTP exhibits the strongest emission in the G-PC gel."*

Supplementary Fig. 33 (Discussion): *"The FL and RTP spectra for B-, G-, and R-PC gels, as well as disordered gel, are measured under identical conditions. As shown in the Supplementary Fig. 33a and b, FL exhibits the strongest intensity in the B-PC gel, while RTP shows the strongest intensity in the G-PC gel. We measure the FL lifetimes of different gels, and the results show that the B-PC gel has the shortest FL lifetime (Supplementary Fig. 33c). This phenomenon is related to the resonant coupling effect of the PCs and the enhancement of the local density of states (LDOS).⁴⁻⁷ Specifically, when the emission bands of FL and RTP align with the PBG of the PCs, the excited photons are localized in the micro-nano structure of the PCs, enhancing the resonance efficiency between the emitted light and the photonic structure. As a result, the photon radiation recombination rate increases, thereby enhancing the emission intensity. Besides, the strong RTP emission in G-PC gel suggests that the resonance interaction between the emission light and photonic structure is independent of the type of molecular transition, whether S_1-S_0 or T_1-S_0 ."*

Fig. R1: **a** FL and **b** RTP spectra of different gels under 365 nm light excitation, θ_1 and θ_2 is 90° . **c** Photoluminescence lifetime curves of different gels excited by a 375 nm laser and monitored at 466 nm wavelength.

References

1. X, Chen et al. Dynamic regulation of photoluminescence based on mechanochromic photonic elastomers. *Chem. Eng. J.* **426**, 131259 (2021).
2. N, Ganesh et al. Enhanced fluorescence emission from quantum dots on a photonic crystal surface. *Nature Nanotech.* **2**, 515-520 (2007).
3. S, Yuan et al. Fluorescence enhancement of perovskite nanocrystals using photonic crystals. *J. Mater. Chem. C* **9**, 908-915 (2021).
4. Lee, H. et al. Structurally engineered colloidal quantum dot phosphor using TiO_2 photonic crystal backbone. *Light: Sci. Appl.* **11**, 318 (2022).

Secondly, it is unclear how the authors can claim that a stronger emission is observed. Although the authors provide evidences in which PL or RTP emission is compared at different observation angles, the increase in emission at certain angles may be due simply to a filter effect brought forth by the angle-dependent PBG. In general, as discussed in the literature regarding PCs (10.1021/cr200063f), the emission of photoactive species found within the PC structure may be influenced by alterations to the local density of states of the overall system that the PC itself causes. In order to prove this point, nonetheless, the authors would need to accurately measure the quantum yield (QY) of their PC system and compare it to

that measured in the absence of PC. If an actual amplification of the emission is found an increase in QY is expected.

Response: We understand the reviewer's concern that the observed enhancement of FL and RTP emission at specific angles in the manuscript may be attributed to the filtering effect caused by the photonic band gap (PBG), rather than actual fluorescence enhancement. We validate the emission enhancement behavior by measuring the absolute photoluminescence quantum yield (PLQY). Considering the inability to measure the PLQY at specific angles, we prepare PC gels with different PBGs by carefully adjusting the diameter of the RTP SiO₂ NPs. These PBGs are made to align as closely as possible with the angle-dependent PBGs of the R-PC gel, simulating the PLQY at different angles (Fig. R2a). Additionally, we measure the PLQY of the gel prepared from the poorly arrangement of RTP SiO₂ NPs as a control experiment. As shown in the (Fig. R2b and Fig. R3), compared to the poorly ordered gel, the PLQY at PBGs of 457 nm and 501 nm (corresponding to the PBG of R-PC gel at $\theta=40^\circ$ and $\theta=50^\circ$) increases by approximately 1.7-fold and 1.5-fold, respectively, confirming the authenticity of the emission enhancement. Therefore, when the emission wavelengths of FL and RTP gradually match the PBG, the resonance efficiency between the excited photons and the photonic structure gradually increases, enhancing the FL and RTP emission. All the data are summarized in Table R1.

In the manuscript, we investigate the angle-dependent chromatic behavior by placing the PC gel between a black light-absorbing substrate and high-transparency quartz glass, and using a black baffle to divide it into two parts: one is the light source irradiation area and the other is the observation and signal detection area (Fig. R2c). The purpose of this setup is to extend the propagation path of the FL and RTP emitted by RTP SiO₂ NPs in the PC gel, thereby enhancing the interaction efficiency between the light and the photonic structure. When the excitation light irradiates the light source area, the RTP SiO₂ NPs generate FL and RTP signals, which propagate within the gel and reach the signal receiving area. Considering that the emission light located in the PBG will be prohibited from propagating, thus, if the emission enhancement is caused by the filter effect, the propagation of the FL and RTP emission bands located in the

PBG within the gel will be inhibited, leading to weaker signals of the PBG-matched band in the receiving area. However, experimental results show that the inherent FL and RTP peak intensities of the three PC gels are much weaker than those of the peaks caused by the PBG, indicating that the angle-dependent chromatic behavior of the PC gels is not caused by the filter effect (Fig. R2d and e). Based on the above discussion, the observed emission enhancement at specific angles is primarily caused by the interaction between light and matter. As shown in the Fig. R2f, the FL and RTP spectra of RTP SiO₂ NPs have wide emission characteristics, covering the entire angle-dependent PBG. When the emission wavelengths of FL and RTP align with the PBG of PCs at specific angles, photons become localized within the crystal due to the bandgap suppression effect. These photons are then guided into specific directions via Bragg reflection, leading to a redistribution of the local density of states, resulting in angle-dependent enhancement of emission intensity.

Fig. R2: **a** Reflection spectra of R-PC gel at different angles (top) and the PC gel prepared with RTP SiO₂ NPs of different particle sizes (bottom). **b** PLQY of the PC gel prepared with RTP SiO₂ NPs of different particle sizes and the poorly ordered gel. **c** Schematic illustration for achieving angle dependent chromatic behavior of PC gels. **d** FL and **e** RTP spectra of B-, G-, and R-PC gels, θ_1 and θ_2 is 90° . **f** The matching situation between the angle-dependent PBG of R-PC gel with the FL and RTP of RTP SiO₂ NPs.

Fig. R3: PLQYs of PC gels prepared by RTP SiO₂ NPs with different diameter.

Table R1: PLQYs of PC gels prepared by RTP SiO₂ NPs with different diameter for simulating the angle-dependent PLQYs of R-PC gel, n=5.

Angle-dependent PBG of R-PC gel		Simulated PC gel		PLQY (%)
Angle (°)	PBG (nm)	Diameter (nm)	PBG (nm)	
90	613	284	613	19.34 ± 0.82
80	590	275	592	21.80 ± 0.76
70	563	271	561	22.12 ± 0.39
60	535	260	538	24.06 ± 0.57
50	501	241	500	25.99 ± 0.64
40	456	217	453	29.35 ± 0.70
30	405	190	406	26.01 ± 0.33
Poorly ordered PC (284 nm diameter)				17.25 ± 0.17

2. When discussing the thermochromic properties of the PC gels, the authors write that because of (a) the overlap between the scattering band of the gels and their own emission band and (b) of the overlap between the excitation band and the emission band, the RTP SiO₂ NPs emission is then scattered in all directions and then continuously transferred, absorbed and re-emitted among particles. This, the authors claim, would thus enhance the PL and RTP intensity (lines 382-388). The supposed explanation is counterintuitive as, unless amplified emission arises, a

stronger re-absorption of photons would lead to a poorer QY. This fact be easily understood by considering the competing non-radiative decay processes which can easily overtake the overall relaxation mechanism after the emitted light is reabsorbed.

Response: We appreciate the reviewers' in-depth consideration of the mechanism behind this scattering-enhanced emission. After careful verification, we sincerely apologize for the inaccurate mechanism. The suggestion that photon reabsorption might lead to a decrease in PLQY is correct. In the manuscript, the prepared RTP SiO₂ NPs exhibit down-conversion luminescence characteristics and excitation-dependent emission feature. Therefore, the RTP SiO₂ NPs emit longer wavelength light upon reabsorption of their own FL or RTP. This behavior would lead to a noticeable redshift in the wavelength of the FL or RTP collected in the low-temperature scattering state (0 °C) compared to the transparent state (40 °C). However, the measured intrinsic FL and RTP emission peaks show no significant shifts (Fig. R4a and b). Consequently, no effective self-absorption behavior between RTP SiO₂ NPs particles occurred in the dynamic thermally induced self-scattering emission enhancement. The almost unchanged PLQY measured at different temperatures confirms that the intrinsic luminescent state of the RTP SiO₂ NPs is unaffected in this process (Fig. R4c), and the emission enhancement mainly originates from the external environment, such as scattered excitation light and temperature. Furthermore, in the manuscript, we observe an increase in the FL lifetime of the R-PC gel at low temperatures (0 °C), which also implies a reduction in non-radiative recombination processes (Fig. R4d). We speculate that the increase in lifetime at 0 °C is primarily due to two factors. Under strong scattering conditions, the incident UV light undergoes multiple reflections and redirections within the PCs, which not only increases the scattering intensity of the UV excitation light but also significantly enhances the interaction between the scattered UV excitation light and RTP SiO₂ NPs. Although the direct absorption of incident UV light by RTP SiO₂ NPs may slightly decrease due to surface scattering enhancement, more scattered UV light will be "captured" inside the PCs and interact repeatedly with the particles. This behavior causes electrons to be continuously refilled into the excited

state, forming a cycle of excitation, decay and re-excitation (S_0 - S_1 - S_0 - S_1), which is equivalent to prolonging the decay time of electrons during the decay process by sustaining repeated energy transitions. On the other hand, low temperatures can suppress molecular vibrations and thereby reduce non-radiative recombination. We have measured the intrinsic temperature-dependent FL lifetime of RTP SiO_2 NPs and found that compared to the significant increase in lifetime caused by self-scattering (from 7.7 ns to 11.8 ns), the intrinsic temperature-dependent FL lifetime remained almost unchanged (from 6.2 ns to 6.6 ns) within the temperature range of 0 °C to 40 °C (Fig. R4e), thus this factor is excluded. Moreover, we have further verified that the photonic structure of the PC gel affects this thermally induced self-scattering emission enhancement behavior. As shown in the Fig. R4f, the poorly ordered gel of RTP SiO_2 NPs exhibits similar enhancement behavior to R-PC gel, indicating that this thermally induced self-scattering enhanced luminescence is mainly achieved by amplifying the excitation light signal in the scattering state.

We have modified the scattering enhancement mechanism and the corresponding mechanism diagram (Fig. R4g) in the manuscript, as well as supplemented the above discussion content to the Supplementary Information (Supplementary Fig. 45 and 48). The revision is highlighted in red in the manuscript and listed below:

On page 18: *"The intrinsic physical mechanism of the thermal-induced self-scattering enhanced FL and RTP properties of the R-PC gel is further investigated. The scattering intensity of UV-excited light collected on the PC gel increases with decreasing temperature, suggesting that the scattering state at low temperatures can amplify the intensity of excitation light, thus leading to stronger photoexcitation of the inside RTP SiO_2 NPs (Supplementary Fig. 43). In other words, the interfacial scattering enhancement firstly induces multiple omni-directional scattering of the incident UV light inside the gel, which improves the overall excitation efficiency of RTP SiO_2 NPs (Fig. 5g).⁴² Although the direct absorption of incident UV light by RTP SiO_2 NPs may slightly decrease due to surface scattering enhancement, more scattered UV light will be "captured" inside the gel and interact repeatedly with the particles. This behavior causes electrons to be continuously refilled into the excited state, forming a cycle of*

excitation, decay and re-excitation, which is equivalent to prolonging the decay time of electrons during the decay process by sustaining repeated energy transitions, leading to an increase in FL lifetime from 7.7 ns in the transparent state to 11.8 ns in the scattering state (Supplementary Fig. 44 and Fig. 45). On the other hand, considering that low temperature limits the thermal motion of molecules in RTP SiO₂ NPs, more energy is released through radiative transitions. Therefore, we use a photo-initiator (2-hydroxy-2-methylpropiophenone) to crosslink the gel, making its optical scattering properties independent of temperature (Supplementary Fig. 46), to investigate whether the fundamental mechanism of luminescence enhancement in R-PC gel during cooling from 40 °C to 0 °C originates from scattering enhancement caused by refractive index mismatching. Upon cooling from 40 °C to 0 °C, the crosslinked PC gel exhibits 1.07-fold and 1.13-fold enhancements in FL and RTP intensities, respectively, accompanied by a 1.04-fold prolongation of RTP lifetime (Supplementary Fig. 47). Therefore, the weak enhancement originating from suppressed molecular thermal motion at low temperatures strongly suggests that the refractive index mismatching in gel systems is the main reason for self-scattering-enhanced luminescence. Furthermore, the thermally-induced self-scattering enhancement behavior is independent of the PC structure, relying mainly on the amplification of the excitation light signal in the scattering state (Supplementary Fig. 48)."

Supplementary Fig. 45 (Discussion): *"To verify that the increase in lifetime shown in Supplementary Fig. 44 is independent of the intrinsic temperature-dependent optical properties of RTP SiO₂ NPs, We have measured the intrinsic temperature-dependent FL lifetime of RTP SiO₂ NPs and found that compared to the significant increase in lifetime caused by self-scattering, the intrinsic temperature-dependent FL lifetime remained almost unchanged within the temperature range of 0 °C to 40 °C (Supplementary Fig. 45), thus the intrinsic temperature-dependent FL lifetime of RTP SiO₂ NPs is excluded."*

Fig. R4: Normalized **a** FL and **b** RTP spectra of R-PC gel measured at different temperatures. **c** PLQY of R-PC gel measured at different temperatures. **d** The FL decay spectra of R-PC gel at 0 °C and 40 °C, respectively. **e** The FL decay spectra of RTP SiO₂ NPs at 0 °C and 40 °C, respectively. **f** The FL and RTP spectra of poorly ordered gel at 0 °C and 40 °C, respectively. **g** A plausible mechanism for thermal-induced self-scattering enhancement of the FL and RTP.

The observed increase in intensity, on the other hand, can be easily explained by the suppression of non-radiative decay pathways which normally occurs by lowering the temperature. These considerations would also explain the increase in emission lifetime observed at lower temperatures. The authors should clarify these points for instance by studying the temperature dependency of the emission efficiency of RTP SiO₂ NPs and then comparing it to that of PC gels. Additionally, in the discussion

the authors claim that temperature-gated self-scattering effects amplify the emission intensity (lines 429-432). In view of the above points, also this claim should be reconsidered unless an amplification of the emission is actually shown.

Response: We thank the reviewer for these important questions. In fact, as the reviewers have pointed out, low temperatures can limit the thermal motion of molecules, thereby causing more energy to be released through radiative transitions. To investigate whether the underlying mechanism of the enhanced luminescence observed in R-PC gel during cooling from 40 °C to 0 °C originates from scattering enhancement induced by refractive index mismatch, the gel is crosslinked using a photo-initiator (2-hydroxy-2-methylpropiophenone) to render their optical scattering properties temperature-independent (Fig. R5a). This experimental design effectively eliminated the interference of temperature-dependent scattering effects on luminescence performance. Upon cooling from 40 °C to 0 °C, the crosslinked PC gels exhibited 1.07-fold and 1.13-fold enhancements in FL and RTP intensities, respectively, accompanied by a 1.04-fold prolongation of RTP lifetime (Fig. R5b, c and d). Notably, these enhancement factors are substantially lower than those observed in non-crosslinked PC gels (128-fold for FL, 87-fold for RTP, and 25-fold for lifetime). The weak enhancement originating from suppressed molecular thermal motion at low temperatures strongly suggests that the refractive index mismatching in gel systems is the main reason for self-scattering-enhanced luminescence. This comparative analysis conclusively validates the feasibility of manipulating luminescence through refractive index-modulated scattering enhancement in photonic crystal materials.

We believe that most readers will question the interference of the inherent temperature effect of fluorophores on the scattering enhancement behavior. To make the manuscript more comprehensive and informative, we have supplemented these research results and detailed discussion in the revised Supplementary Information (Supplementary Fig. 46 and 47). The revision is highlighted in red in the manuscript and listed below:

On page 19: *"On the other hand, considering that low temperature limits the thermal motion of molecules in RTP SiO₂ NPs, more energy is released through*

radiative transitions. Therefore, we use a photo-initiator (2-hydroxy-2-methylpropiophenone) to crosslink the gel, making its optical scattering properties independent of temperature (Supplementary Fig. 46), to investigate whether the fundamental mechanism of luminescence enhancement in R-PC gel during cooling from 40 °C to 0 °C originates from scattering enhancement caused by refractive index mismatching. Upon cooling from 40 °C to 0 °C, the crosslinked PC gel exhibits 1.07-fold and 1.13-fold enhancements in FL and RTP intensities, respectively, accompanied by a 1.04-fold prolongation of RTP lifetime (Supplementary Fig. 47). Therefore, the weak enhancement originating from suppressed molecular thermal motion at low temperatures strongly suggests that the refractive index mismatching in gel systems is the main reason for self-scattering-enhanced luminescence."

Fig. R5: **a** Transmission spectra of cured R-PC gel with photo-initiators under different temperature. **b** FL, **c** RTP and **d** RTP decay spectra of cured R-PC gel under different temperature.

3. Fundamental to the importance of this work is the mass-production capabilities of the developed synthesis. The authors claim that more than 700 g can be obtained in a single batch. Nonetheless, the synthetic yield of the synthesis (both at low and large scale) should be reported to put the resulting mass-production capabilities into perspective.

Response: We are grateful to the reviewer for the constructive suggestions on the large-scale preparation of this work. In order to more intuitively and comprehensively demonstrate the capabilities of designed strategy for the large-scale preparation of self-assembled monodisperse RTP SiO₂ NPs, we conducted 25-fold and 250-fold proportional amplification experiments on the basis of the original preparation experiments (1-fold reference amount) for verification. As presented in Fig. R6, through the two preparation steps of Stöber method and calcination, the RTP SiO₂ NPs obtained at three different preparation scales all exhibit FL and RTP. Importantly, their morphology and self-assembly capability remain unaffected by the scaling of reaction precursors, demonstrating excellent scalability of our designed synthesis strategy. Notably, thanks to the high flexibility of the Stöber method for production equipment, we have achieved laboratory-scale preparation at the hundred-gram level (700 g/batch). Furthermore, we use the product mass obtained from the original synthetic method (1-fold) as the criterion to evaluate the yield of large-scale preparation. Table R2 demonstrates that even when scaling up the reaction system by 250-fold, the yield remains consistently above 99.2% with a batch-to-batch variation of less than 0.8%. These results strongly validate the exceptional scalability and process stability of this synthetic methodology.

To enhance the research completeness of large-scale preparation in the manuscript, we have supplemented these research results and detailed discussion in the revised Supplementary Information (Supplementary Fig. 31 and Supplementary Table 4).

Supplementary Fig. 31 (Discussion): *"In order to more intuitively and comprehensively demonstrate the capabilities of designed strategy for the large-scale preparation of self-assembled monodisperse RTP SiO₂ NPs, we conducted 25-fold and 250-fold proportional amplification experiments on the basis of the original preparation experiments (1-fold reference amount) for verification. As presented in Supplementary Fig. 31, through the two preparation steps of Stöber method and high-temperature calcination, the RTP SiO₂ NPs obtained at three different preparation scales all exhibit FL and RTP. Importantly, their morphology and self-assembly capability remain unaffected by the scaling of reaction precursors, demonstrating*

excellent scalability of our designed synthesis strategy. Notably, thanks to the high flexibility of the Stöber method for production equipment, we have achieved laboratory-scale preparation at the hundred-gram level (700 g/batch). Furthermore, we use the product mass obtained from the original synthetic method (1-fold) as the criterion to evaluate the yield of large-scale preparation. Supplementary Table 4 demonstrates that even when scaling up the reaction system by 250-fold, the yield remains consistently above 99.2% with a batch-to-batch variation of less than 0.8%. These results strongly validate the exceptional scalability and process stability of this synthetic methodology."

Fig. R6: a-c Photograph of RTP SiO₂ NPs prepared at magnifications of 1-fold, 25-fold and 250-fold. d-f The corresponding photographs of large-scale prepared RTP SiO₂ NPs under sunlight, UV radiation and after UV shut-off. g-i SEM images of RTP SiO₂ NPs prepared at different scales after self-assembly.

Table R2: The mass and yield of RTP SiO₂ NPs prepared at different scales.

Magnification factor	Precursor solution (mL)				Mass (g)	yield
	Ethanol	Ammonia	Glucose solution	TEOS/Ethanol		
1-fold	130	16	6	12/10	2.9±0.03	100%
25-fold	3250	400	150	300/250	72.1±0.93	~99.4%
250-fold	32500	4000	1500	3000/2500	719.4±2.08	~99.2%

4. In order to prove that the embedding of CDs inside SiO₂ NPs can enhance the RTP thanks to the stabilizing role of C-Si bonds, the authors report that the RTP decay lifetime is the longest when calcination is performed at a temperature of 575 °C. This point is quite important as it further proves that the SiO₂ structure can effectively stabilize the CDs triplet state by suppressing non-radiative recombination pathways and leading to an enhanced RTP. Nonetheless, despite the different signal-to-noise ratio, the RTP decay traces reported in the Supplementary Fig. 17 all proceed parallel to each other in a semi-logarithmic plot. If, as reported by the authors, the RTP lifetime varies between 2s and 0.1s, the slope of the traces should vary of factor of 20 in such plot and not stay parallel. The authors do not discuss how the time-traces have been analyzed and the reported lifetimes have been obtained. The least-squares fitting procedure and the model functions utilized to fit the experimental data should be discussed in detail to clarify this point. If a multiexponential decay has been used as model function, the lifetime and amplitudes of each single exponential should be reported, at least in the supplementary material. Additionally, the confidence interval of all parameters obtained from the fits should be reported.

Response: Thank you very much for your careful reading. After careful review, as the reviewer pointed out, the trajectories of these RTP decay curves are almost parallel. These data are measured in the dynamic decay mode in Fluoracle[®] software. In detailed, the total measurement time for each decay curve is maintained at 50 s. Firstly, RTP SiO₂

NPs powder obtained in the calcination temperature range from 325 °C to 825 °C is continuously radiated by a xenon lamp (365 nm) for 25 s, and then the RTP decay curve at 504 nm is collected within 25 s to 50 s after the xenon lamp is turned off. All the lifetime decay curves are fitted based on the tri-exponential equation, and the average lifetime (τ_{av}) is calculated according to the following equation:

$$\tau_{av} = \frac{\sum A_i \tau_i^2}{\sum A_i \tau_i}$$

Unlike the microsecond lamp measurement method, the dynamic decay mode requires precise optimization of the excitation energy of xenon lamp to achieve saturation of triplet excitons in the sample, thereby enabling the acquisition of reliable RTP decay curves. Based on this, we speculate that the raw measured RTP decay curve might be due to the insufficient excitation energy provided by the xenon lamp, which prevented the accurate collection of the rapidly decay components during the signal collection stage, resulting in the curves presenting a similar slowly decay signal.

Therefore, we increased the excitation intensity of the xenon lamp and re-measured the relevant data. As shown in Fig. R7a-c, when the calcination temperature increases from 325 °C to 575 °C, the inclination of the RTP decay curve gradually slows down and the corresponding τ_{av} increases from 0.47 s to 2.02 s. As the calcination temperature further increases to 825 °C, the inclination becomes steeper rapidly and the τ_{av} decreases to 0.49 s (825 °C). The relationship between the remeasured τ_{av} and the calcination temperature follows a consistent pattern with the raw measurement results. To further confirm the reliability of the data, we re-prepared RTP SiO₂ NPs at different calcination temperatures and measured their RTP decay curve under the same method. As presented in Fig. R7d-f, both the RTP decay curve and the fitted τ_{av} have similar results, further confirming the accuracy and reliability of the collected data. Besides, we have summarized the fitted lifetime parameters for RTP at different calcination temperatures, as shown in Table R3, and all quantitative measurements are performed in five replications (n=5) to calculate 95% confidence intervals. Additionally, all fitting parameters for the temperature-dependent RTP lifetimes (77 K-347 K) are also determined and presented in Table R4.

Based on the above experimental results, we have modified the RTP decay curve of RTP SiO₂ NPs prepared at different calcination temperatures (Supplementary Fig. 18). All fitting parameters are supplemented in the Supplementary Table 1 and Table 2. Besides, we have also added detailed RTP and FL lifetime measurement and fitting methods to enhance the completeness of the manuscript:

Supplementary method (Measurement and fitting of RTP and FL lifetime):

"The phosphorescence lifetime measurements are performed using the dynamic decay mode integrated into the Fluoracle® software of Edinburgh FLS1000 fluorescence spectrometer. Each decay curve was acquired over a total acquisition time of 50 s. Specifically, these samples, including RTP SiO₂ NPs powder, self-assembled PCs and PC gels, are subjected to continuous irradiation with a 365 nm xenon lamp for 25 s. Following irradiation cessation, time-resolved RTP decay profiles at 504 nm are recorded during the subsequent 25-50 s interval. The acquired decay curves are analyzed via tri-exponential fitting using the equation:

$$I(t) = A_1 \exp(-t/\tau_1) + A_2 \exp(-t/\tau_2) + A_3 \exp(-t/\tau_3)$$

Where $I(t)$ represents the phosphorescence intensity at time t . A_i denotes the pre-exponential factors, and τ_i corresponds to the decay time constants. The average lifetime (τ_{av}) is subsequently calculated using the expression:

$$\tau_{av} = \sum A_i \tau_i^2 / \sum A_i \tau_i$$

Besides, the measurement of FL lifetime is conducted using a 375 nm laser monitored at 466 nm, and analysis protocols for decay curves are identical with RTP lifetime."

Fig. R7: **a** and **b** The re-measured RTP lifetime decay spectra of RTP SiO₂ NPs obtained at different calcination temperatures. **c** The relationship between RTP lifetime and calcination temperature. **d** and **e** The RTP lifetime decay spectra of re-prepared RTP SiO₂ NPs obtained at different calcination temperatures. **f** The relationship between RTP lifetime of re-prepared RTP SiO₂ NPs and calcination temperature.

Table R3: Summary of RTP lifetimes of the RTP SiO₂ NPs obtained at different calcination temperatures, n=5.

T (°C)	τ_1 (s)	A ₁	τ_2 (s)	A ₂	τ_3 (s)	A ₃	τ_{av} (s)
325	0.247± 0.054	246± 191	0.035± 0.009	499± 13	1.219± 0.060	24± 1.6	0.473± 0.029
375	0.181± 0.064	285± 12	0.377± 0.011	208± 8	1.298± 0.036	118± 9	0.843± 0.023
425	0.162± 0.031	1512± 65	0.734± 0.026	824± 39	2.008± 0.078	190± 11	1.041± 0.091
475	0.572± 0.026	1396± 68	1.159± 0.115	1611± 67	2.976± 0.11	309± 4	1.499± 0.030
525	2.283± 0.052	1781± 54	1.286± 0.030	1838± 57	3.323± 0.084	351± 11	1.753± 0.066
575	0.660± 0.053	2337± 61	1.977± 0.068	1845± 57	3.856± 0.112	317± 14	2.021± 0.036
625	0.445± 0.040	1494± 51	1.573± 0.043	1411± 33	3.053± 0.045	315± 19	1.749± 0.039
675	0.137± 0.034	1232± 111	0.631± 0.026	1296± 116	1.889± 0.088	433± 16	1.156± 0.073
725	0.362± 0.041	602± 35	0.102± 0.010	473± 21	1.339± 0.078	153± 9	0.773± 0.053
775	0.207± 0.051	294± 15	0.073± 0.004	339± 17	1.191± 0.051	58± 2	0.631± 0.031
825	0.303± 0.048	284± 18	0.223± 0.016	224± 21	0.831± 0.079	103± 4	0.491± 0.043

Table R4: Summary of RTP lifetimes of the RTP SiO₂ NPs obtained at different temperatures (77 K-347 K), n=5.

T (K)	τ_1 (s)	A ₁	τ_2 (s)	A ₂	τ_3 (s)	A ₃	τ_{av} (s)
77	1.328± 0.045	3720± 111	1.278± 0.065	5219± 132	11.591± 0.484	420± 33	4.378± 0.362
107	1.206± 0.063	3141± 84	1.555± 0.050	1239± 76	8.147± 0.701	377± 11	3.760± 0.349
137	1.079± 0.057	3152± 47	2.827± 0.058	1175± 52	7.221± 0.386	355± 11	3.404± 0.144
167	0.804± 0.035	4255± 153	3.030± 0.045	1726± 64	7.081± 0.336	314± 8	3.162± 0.116
197	0.852± 0.022	4730± 92	3.108± 0.104	2429± 102	6.710± 0.494	335± 31	3.033± 0.712
227	0.582± 0.026	4405± 50	2.506± 0.129	1637± 68	5.931± 0.429	370± 9	2.731± 0.235
257	0.753± 0.053	4935± 56	2.739± 0.044	2715± 58	5.216± 0.383	329± 6	2.498± 0.098
287	0.726± 0.047	3237± 96	2.457± 0.118	1992± 68	4.230± 0.331	313± 5	2.258± 0.114
317	0.583± 0.043	1318± 29	1.937± 0.081	1412± 78	3.120± 0.227	260± 6	1.920± 0.097
347	0.475± 0.024	1167± 44	1.710± 0.073	1497± 43	2.806± 0.233	241± 7	1.728± 0.070

5. No details regarding the QY calculations are provided, although QY values are reported in Supplementary Fig. 19 and used to discuss the role in the SiO₂ structure in the enhancement of the CDs emission. The authors should discuss in detail the method used to obtain the absolute QYs values in the Methods section or in the supplementary material. Additionally, by observing Supplementary Fig. 16, it appears that at calcination temperatures higher than 575 °C the QY of the system decreases. Why are QYs of the compound obtained at these calcination temperature not reported? Since (a) crystallinity of the system increases with calcination temperature even above 575 °C (lines 197-202) and (b) the authors claim that the crystallinity helps to enhance the QY (lines 193-196), the authors should discuss what leads to the QY reduction above 575 °C. Finally, the confidence intervals associated to the reported QYs values should be indicated as well.

Response: Thanks for the reviewer’s insightful questions and suggestion. The absolute quantum yields (PLQY) in the manuscript are measured based on the Edinburgh FLS1000 fluorescence spectrometer equipped with an integrating sphere accessory. In detail, the background spectra of the excitation light at 365 nm is scanned first, the scanning steps are 0.3 nm and the range is 345-800 nm. Subsequently, the RTP SiO₂ NPs obtained at different calcination temperatures are placed on the sample platform inside the integrating sphere, and the mass of the RTP SiO₂ NPs powder is guaranteed to be 0.07 g. Then, the photoluminescence spectra of these RTP SiO₂ NPs are recorded under the same conditions. The calculation of PLQY is accomplished through the quantum yield analysis module of the Fluoracle[®] software built into the Edinburgh FLS 1000 system. The scattering region of the excitation light (~350-378 nm) and the fluorescence emission region of the sample (~378-780 nm) are respectively set as integral intervals, and the values are automatically calculated in combination with the absolute quantum yield equation:

$$PLQY = \frac{\textit{Emitted photons}}{\textit{Absorbed photons}} = \frac{\int L_{\textit{sample}}(\lambda)d\lambda}{\int [E_{\textit{ref}}(\lambda) - E_{\textit{sample}}(\lambda)]d\lambda}$$

Where the L_{sample} represents the emission spectra of the sample, and E_{ref} and E_{sample} are the excitation luminous fluxes without and with the sample, respectively.

Based on the above method, we additionally measured the PLQYs of RTP SiO₂ NPs obtained in the calcination temperature range from 625 °C to 825 °C, as shown in the Fig. R8 and Table R4. As the calcination temperature increases from 625 °C to 825 °C, the QYs of RTP SiO₂ NPs gradually decreases. In order to clarify the reasons for the reduction of QYs caused by excessive high-temperature calcination, we conducted a systematic exploration. As the calcination temperature increases from 625 °C to 825 °C, the intensity ratio (I_G/I_D) of crystalline G band and disordered D band in Raman spectra shows a trend of increasing first and then decreasing (Fig. R9a). Combining the Raman spectra measured in the range of 325 °C -575 °C for summary, we found that I_G/I_D shows a non-monotonic changing trend, gradually increasing in the range of 325-675 °C, reaching the peak at 675 °C, and then significantly decreasing in the range of 675-825 °C (Fig. R9b). Therefore, we speculate that the influence of calcination

temperature on the in-situ formation of CDs inside SiO₂ can be divided into three intervals: in the first stage, as the temperature increases from 325 °C to 575 °C, the organic small molecules SiO₂ undergo carbonization and crystallization to form CDs, which has the optimal optical properties at the calcination temperature of 575 °C. The CDs generated at 575 °C has a high crystallinity and a rich variety of chemical defects (O-related defects) on the surface, which can enhance the ability to capture electrons and provide additional radiative recombination transition channels for electrons. As the temperature further increases (from 575 °C to 675 °C), the crystallinity of CDs will further increase, but the chemical defects are damaged due to the high temperature, thus the luminous efficiency begins to decrease. However, when the temperature further increases and exceeds 675 °C, the structure of CDs is cracked by high temperature, resulting in a rapid decline in optical performance.

To verify this hypothesis, we have measured the UV-Vis absorption spectra and XPS spectra of RTP SiO₂ NPs generated at calcination temperatures ranging from 625 °C to 825 °C. As shown in Fig. R9c, as the calcination temperature increases, the broad absorption band in the range of approximately 350-500 nm, attributed to the n-π* electronic transitions of surface defect states in the CDs, gradually diminishes. This indicates that increased temperatures lead to a reduction in the chemical defect states on the surfaces of CDs. Additionally, for RTP SiO₂ NPs calcined at 775 °C and 825 °C, the high-energy π-π* electronic transition absorption peak at ~258 nm, originating from the carbon cores of the CDs, is significantly weakened, suggesting structural degradation of the carbon cores. Furthermore, the fitted high-resolution C 1s results of RTP SiO₂ NPs generated at calcination temperatures shown in Fig. R9d demonstrate that they all contain three peaks at about 282.2 eV, 285.0 eV and 288.5 eV, which are assigned to C-Si, C-C/C=C and C=O/C-O, respectively. The fitting results show an increase in the relative content of C-Si bonds and a decrease in C-C/C=C and C=O/C-O bonds. Although the content of C-Si continues to increase, the excessively high calcination temperature will lead to the destruction of the chemical defect state, and cause the partial cleavage of the CDs, which ultimately leading to a decrease in the overall optical performance. The corresponding high-resolution Si 2p and O 1s fitting

results further confirm this view (Fig. R9e-f). In conclusion, based on the above discussion, it can be inferred that when the calcination temperature exceeds 625 °C, the decrease in PLQY of RTP SiO₂ NPs is mainly due to the reduction of chemical defect states of internal CDs and the cleavage of their own structure (Fig. R9g).

As suggested, we have supplemented the measurement method of PLQY in the Characterization of the experimental section, and the PLQY data for RTP NPs obtained at 625-825 °C calcination are included in the Supplementary Information (Supplementary Fig. 20 and Supplementary Table 3). Besides We have also supplemented the above discussion in the Supplementary Fig. 27 and Supplementary method. The revision is highlighted in red in the manuscript and listed below:

On page 24: *"The PLQY is measured based on the Edinburgh FLS1000 fluorescence spectrometer equipped with an integrating sphere accessory, the scanning steps are 0.3 nm and the range is 345-800 nm."*

Supplementary method (Measurement and calculation of PLQY): *"The PLQY are measured based on the Edinburgh FLS1000 fluorescence spectrometer equipped with an integrating sphere accessory. In detail, the background spectra of the excitation light at 365 nm are scanned first, the scanning steps are 0.3 nm and the range is 345-850 nm. Subsequently, the samples are placed on the sample platform inside the integrating sphere, the mass of the powder samples is guaranteed to be 0.07 g, and the volume of the liquid gel samples is 1 mL. Then, the photoluminescence spectra of these RTP SiO₂ NPs are recorded under the same conditions. The calculation of PLQY is accomplished through the quantum yield analysis module of the Fluoracle[®] software built into the Edinburgh FLS 1000 system. The scattering region of the excitation light (~350-378 nm) and the fluorescence emission region of the sample (~378-780 nm) are respectively set as integral intervals, and the values are automatically calculated in combination with the absolute quantum yield equation:*

$$PLQY = \frac{\text{Emitted photons}}{\text{Absorbed photons}} = \frac{\int L_{\text{sample}}(\lambda)d\lambda}{\int [E_{\text{ref}}(\lambda) - E_{\text{sample}}(\lambda)]d\lambda}$$

Where L_{sample} represents the emission spectra of the sample, and E_{ref} and E_{sample} are the excitation luminous fluxes without and with the sample, respectively."

Fig. R8: PLQYs of RTP SiO₂ NPs obtained at different calcination temperatures.

Fig. R9: **a** The Raman spectra of RTP SiO₂ NPs obtained at different calcination temperatures. **b** The relationship between I_G/I_D and calcination temperatures. **c** UV-vis absorption spectra of RTP SiO₂ NPs obtained at different calcination temperatures. The high-resolution **(d)** Si 2p, **(e)** C 1s and **(f)** O 1s results of RTP SiO₂ NPs obtained at different calcination temperatures. **g** A plausible mechanism for the PLQY decrease in RTP SiO₂ NPs obtained under excessive temperature calcination.

Table R4: Summary of the absolute photoluminescence quantum yields (PLQY) of RTP SiO₂ NPs obtained at different calcination temperatures, n=5.

T (°C)	PLQY (%)
325	6.10 ± 0.29
375	8.40 ± 0.17
425	9.45 ± 0.31
475	11.29 ± 0.55
525	15.76 ± 0.21
575	18.13 ± 0.60
625	15.63 ± 0.34
675	11.32 ± 0.26
725	10.89 ± 0.74
775	10.05 ± 0.53
825	7.50 ± 0.58

In addition to the major points discussed above, the following minor points should be addressed as well:

6. In the discussion section (lines 420-423) the authors claim that the covalent Si-C network stabilizes the triplet states through spatial confinement. The term spatial confinement suggests some sort of quantum confinement effect. Such quantum effect is not discussed in the work while, as also discussed by the authors, the Si-C network helps in the reduction of the non-radiative processes competing with the radiative recombination leading to RTP. I suggest the authors to rephrase the parts of the text where the term confinement is used to make the discussion clearer and avoid confusion.

Response: We thank the reviewer for the careful review. We have revised the relevant expressions according to the suggestions. The revision is highlighted in red in the manuscript and listed below:

On page 2: *"During the calcination process, the embedded organic molecules in SiO₂ matrix undergo in situ formation of fluorescent carbon dots (FL CDs) through carbonization, aggregation and crystallization, while the covalent C-Si bond network between the CDs and SiO₂ matrix is gradually build up, stabilizing the triplet excited state to produce long-lived RTP emission."*

On page 22: *"During high-temperature calcination, the introduced organic small molecules embedded in silica networks undergo in situ carbonization and crystallization to form CDs, while the rigid environment provided by the C-Si covalent bond efficiently suppresses non-radiative transitions and facilitates the stabilization of triplet excitons, enabling robust RTP emission."*

7. At lines 132, 208 and 434 the authors use the term chemiluminescence to discuss the optical properties of their system. Nonetheless, the emission explored in the manuscript does not originates from chemiluminescent processes. In such kind of processes the energy leading to the emission is provided by chemical reactions. Differently, in this work the emission always originates from photoexcitation. I suggest the authors to rephrase all claims of chemiluminescence.

Response: Thanks for your kind reminder. In theory, photoluminescence (PL) includes short-lifetime fluorescence (FL) and long-lifetime phosphorescence, as both of them belong to the process in which materials absorb the energy of photons and then re-emit light. Therefore, we have modified the chemiluminescence in the manuscript to photoluminescence. The revision is highlighted in red in the manuscript and listed below:

On page 4: *"Clearly, by combining physical structures with photoluminescence, our findings provide a feasible approach for constructing optical devices with multi-stimulus response and customized optical signal expression."*

On page 6: *"In addition, the optical intensity of RTP SiO₂ NPs also maintains excellent stability even in different organic solvents and metal ion solutions (Supplementary Fig. 8 and Fig. 9), which reflects a good anti-quenching ability, and*

further proves that the photoluminescence center of RTP SiO₂ NPs is located in the interior of SiO₂ NPs."

On page 10: *"Consequently, the photoluminescence centers of RTP SiO₂ NPs after calcination are derived from internal in-situ generated CDs, and further confirms that even a single organic small-molecule carbon source can also be converted to CDs through carbonation, aggregation and crystallization within the silica matrix (Fig. 3d)."*

On page 22: *"The strategy of large-scale preparation of monodisperse RTP SiO₂ NPs and the successful integration of physical photonic structure and photoluminescence provide new approach for constructing advanced multi-mode luminescent devices."*

8. At lines 130-133, the authors claim that the RTP SiO₂ NPs present good anti-quenching ability. Nonetheless in Supplementary Fig. 8 it results that Fe³⁺ ions are able to quench both RTP and PL of the system. The authors should discuss this point.

Response: We sincerely thank the reviewer's patient reading and useful comments on our manuscript. In order to comprehensive understanding of the optical stability of the prepared RTP SiO₂ NPs, the FL and RTP intensity for RTP SiO₂ NPs is investigated in the presence of different organic solvents and different kinds of metal ionic interferences with the concentration of 1 M under identical conditions. It is obvious that Fe³⁺ and Co²⁺ can induce FL and RTP quenching for RTP SiO₂ NPs, while other environments or interfering substances show almost no quenching as demonstrated in Fig. R10a-d. Meanwhile, Fe³⁺ and Co²⁺ also have a significant impact on the RTP lifetime of RTP SiO₂ NPs (Fig. R10e and f). It is worth mentioning that RTP SiO₂ NPs also exhibit a short RTP lifetime in methanol, which is due to low dispersity of RTP SiO₂ NPs in methanol (Fig. R10g). Generally, the quenching of fluorophores by metal ions mainly involves static quenching, dynamic quenching and inner filter effect.¹ Firstly, the FT-IR spectra of RTP SiO₂ NPs in the presence and absence of Fe³⁺ and Co²⁺ show that no obvious shifts and no new vibration absorption sites appear (Fig. R11a), indicating that no functional groups are formed or adhered to the surface of RTP

SiO₂ NPs, thus excluding static quenching.² To seek the reason for the quenching phenomenon, the fluorescence lifetime of RTP SiO₂ NPs with and without Fe³⁺ and Co²⁺ are measured. As shown in Fig. R11b, the fluorescence lifetime of RTP SiO₂ NPs in the absence and presence of Fe³⁺ and Co²⁺ are almost unchanged. The inner filter effect typically requires that the absorption band of the quencher overlap with the excitation or emission band of the fluorophore, resulting in a bursting behavior by absorbing the excitation or emission light of the fluorophore.³ As shown in the Fig. R11c and d, the absorption spectra of Fe³⁺ and Co²⁺ ions overlap with the excitation and RTP emission peaks of RTP SiO₂ NPs, respectively, indicating a typical inner filter effect. Furthermore, the RTP SiO₂ NPs after separation with Fe³⁺ and Co²⁺ solutions exhibit reversible recovery of optical properties (Fig. R11e and f).

To make the manuscript more comprehensive and informative, we have supplemented the research results in the revised Supplementary Information (Supplementary Fig. 9). The revision is highlighted in red in the manuscript and listed below:

On page 6: *"In addition, the optical intensity of RTP SiO₂ exhibits good stability in various organic solvents and most metal ion solutions (Supplementary Fig. 8 and Fig. 9)."*

Supplementary Fig. 9 (Discussion): *"Systematic exploration of the optical stability of RTP SiO₂ NPs in the presence of different organic solvents and different kinds of metal ionic interferences with the concentration of 1 M under identical conditions. Supplementary Fig. 9a-d show the Fe³⁺ and Co²⁺ can induce FL and RTP quenching for RTP SiO₂ NPs. Meanwhile, Fe³⁺ and Co²⁺ also have a significant impact on the RTP lifetime of RTP SiO₂ NPs (Supplementary Fig. 9e and f). It is worth mentioning that RTP SiO₂ NPs also exhibit a short RTP lifetime in methanol, which is due to low dispersity of RTP SiO₂ NPs in methanol (Supplementary Fig. 9g). Generally, the quenching of fluorophores by metal ions mainly involves static quenching, dynamic quenching and inner filter effect.¹ Firstly, the FT-IR spectra of RTP SiO₂ NPs in the presence and absence of Fe³⁺ and Co²⁺ show that no obvious shifts and no new vibration absorption sites appear (Supplementary Fig. 9h), indicating that no*

functional groups are formed or adhered to the surface of RTP SiO₂ NPs, thus excluding static quenching.² To seek the reason for the quenching phenomenon, the fluorescence lifetime of RTP SiO₂ NPs with and without Fe³⁺ and Co²⁺ are measured. As shown in Supplementary Fig. 9i, the fluorescence lifetime of RTP SiO₂ NPs in the absence and presence Fe³⁺ and Co²⁺ are almost unchanged. The inner filter effect typically requires that the absorption band of the quencher overlap with the excitation or emission band of the fluorophore, resulting in a bursting behavior by absorbing the excitation or emission light of the fluorophore.³ As shown in the Supplementary Fig. 9j and k, the absorption spectra of Fe³⁺ and Co²⁺ ions overlap with the excitation and RTP emission peaks of RTP SiO₂ NPs, respectively, indicating a typical inner filter effect."

Fig. R10: FL spectra of RTP SiO₂ NPs dispersed in (a) different solvents and (b) various metal ion solutions. RTP spectra of RTP SiO₂ NPs dispersed in (c) different solvents and (d) various metal ion solutions. RTP decay curve of RTP SiO₂ NPs dispersed in (e) different solvents and (f) various metal ion solutions. g The optical stability of the summarized RTP SiO₂ NPs in different environments, where the I₀ and I represent the emission intensity of RTP SiO₂ NPs dispersed in water and in different environments, respectively.

Fig. R11: **a** FTIR spectra of RTP SiO₂ NPs water dispersion in the presence and absence of Fe³⁺ and Co²⁺. **b** FL decay profile of RTP SiO₂ NPs water dispersion in the presence and absence of Fe³⁺ and Co²⁺. **c** The UV-vis absorption spectra of Fe³⁺, and the FLE and FL spectra of the RTP SiO₂ NPs water dispersion. **d** The UV-vis absorption spectra of Co²⁺, and RTP spectra of the RTP SiO₂ NPs water dispersion. **e** FL and RTP spectra of mixed solutions of RTP SiO₂ NPs with metal ions (Fe³⁺ or Co²⁺) before and after separation. **f** RTP decay curve of mixed solutions of RTP SiO₂ NPs with metal ions (Fe³⁺ or Co²⁺) before and after separation.

References

1. Vervalde, A. M. et al. Quenching of Photoluminescence of Carbon Dots by Metal Cations in Water: Estimation of Contributions of Different Mechanisms. *J. Phys. Chem. C* **44**, 21617-21628 (2023).
2. Wang, C. et al. Synthesis of multi-color fluorine and nitrogen co-doped graphene quantum dots for use in tetracycline detection, colorful solid fluorescent ink, and film. *J. Colloid Interf. Sci.* **602**, 689-698 (2021).

3. Chen, S. et al. Inner filter effect-based fluorescent sensing systems: A review. *Anal. Chim. Acta* **999**, 13-26 (2018).

9. It is unclear what the difference between the panels (f) and (g) of Fig. 2 is and how the two TEM images have been obtained.

Response: Thank you very much for your careful reading. To characterize the assembly behavior of RTP SiO₂ NPs, TEM is employed to visually resolve their packing structures. Fig. R12a and b (Fig 2f and g in manuscript) present the RTP SiO₂ NPs in a double-layer and triple-layer close-packed structure, respectively. In Fig. R12a, the second layer of RTP SiO₂ NPs is clearly observed to nest within the voids formed by the first-layer close-packed spheres, which is a characteristic arrangement with obvious gaps between two layers. By contrast, there are no visible gaps in the three-layer assembly (Fig. R12b), as the NPs in the third layer occupy the gaps created between the first two layers. This results in a stacking sequence where the third layer adopts a distinct stacking form relative to the first two layers, i.e., ABC-stacking, manifesting a typical face-centered cubic (fcc) packing form (Fig. R12c). Therefore, Fig. 2f and Fig. 2g in the manuscript present the close-packed plane of fcc ((111) plane), while Fig. R12d and e show the (100) plane of fcc.

Besides, for the TEM images of RTP SiO₂ NPs in the manuscript, we used copper mesh as the imaging carrier. In detail, the RTP SiO₂ NPs uniformly disperse in the ethanol solution, and the RTP SiO₂ NPs suspension (0.1wt%) was prepared through 20 min of ultrasonic treatment. Then, use a pipette to draw the suspension (5 μL) and drop it onto a copper mesh, and dry it at an ambient temperature of 60 °C for 30 min. As presented in Fig. R12f and g, the surface of the prepared copper mesh can clearly present the structural color of photonic crystals self-assembled from RTP SiO₂ NPs (220 nm diameter).

Fig. R12: **a** and **b** TEM images of RTP SiO₂ NPs in the self-assembled state. **c** Schematic of the stacking of RTP SiO₂ NPs. **d** TEM and **e** SEM images of (100) plane of RTP SiO₂ NPs stacked in face-centered cubes. **f** and **g** The photograph of the copper mesh before and after dripping by RTP SiO₂ NPs suspension.

10. At line 472, regarding the preparation of PC gels, a typo is present. The sentence "to ethanol the alcohol" should be corrected.

Response: Thanks for your kind reminder. After careful review of the manuscript, we have corrected this error in the revised manuscript as follows:

On page 24: *"The PC gel is obtained by evaporating the mixture at 90 °C for 2 hours to evaporate the ethanol, and the volume fraction of RTP SiO₂ NPs, ETPTA, PEGDA and EG in the PC gel is 30%, 35%, 17.5% and 17.5%, respectively."*